# An alternatively spliced, non-signaling insulin receptor modulates insulin sensitivity via insulin peptide sequestration in *C. elegans*

Bryan A Martinez[1†], Pedro Reis Rodrigues[1†‡], Ricardo M Nuñez Medina[1], Prosenjit Mondal[1§], Neale J Harrison[1#], Museer A Lone[1¶], Amanda Webster[1], Aditi U Gurkar[1**], Brock Grill[2], Matthew S Gill[1*]

[1]Department of Molecular Medicine, The Scripps Research Institute – Scripps Florida, Jupiter, United States; [2]Department of Neuroscience, The Scripps Research Institute – Scripps Florida, Jupiter, United States

*For correspondence:
mgill@scripps.edu

†These authors contributed equally to this work

Present address: ‡Department of Chemistry and Chemical Biology, Boyce Thompson Institute, Cornell University, Ithaca, United States; §School of Basic Sciences, BioX, Indian Institute of Technology, Mandi, India; #School of Biosciences, College of Life and Environmental Sciences, University of Birmingham, Birmingham, United Kingdom; ¶Institute for Clinical Chemistry, University Hospital Zurich, Zurich, Switzerland; **Aging Institute, Division of Geriatric Medicine, Department of Medicine, University of Pittsburgh, Pittsburgh, United States

Competing interests: The authors declare that no competing interests exist.

**Abstract** In the nematode *C. elegans*, insulin signaling regulates development and aging in response to the secretion of numerous insulin peptides. Here, we describe a novel, non-signaling isoform of the nematode insulin receptor (IR), DAF-2B, that modulates insulin signaling by sequestration of insulin peptides. DAF-2B arises via alternative splicing and retains the extracellular ligand binding domain but lacks the intracellular signaling domain. A *daf-2b* splicing reporter revealed active regulation of this transcript through development, particularly in the dauer larva, a diapause stage associated with longevity. CRISPR knock-in of mScarlet into the *daf-2b* genomic locus confirmed that DAF-2B is expressed in vivo and is likely secreted. Genetic studies indicate that DAF-2B influences dauer entry, dauer recovery and adult lifespan by altering insulin sensitivity according to the prevailing insulin milieu. Thus, in *C. elegans* alternative splicing at the *daf-2* locus generates a truncated IR that fine-tunes insulin signaling in response to the environment.

## Introduction

Alternative splicing of messenger RNA provides a mechanism for generating protein diversity, and it has been suggested that up to 95% of the human genome undergoes some form of alternative splicing (*Chen and Manley, 2009*). Indeed, it is well established that alternative splicing at the insulin receptor (IR) locus in mammals leads to the expression of two isoforms of the IR that differ in their affinity for ligands (*Belfiore et al., 2009*). There is also evidence in mammals and insects for the existence of alternatively spliced, short isoforms of the IR (*Västermark et al., 2013*; *Vorlová et al., 2011*) but their functional significance remains unknown. Recent studies suggest that dysregulation of alternative splicing is not only associated with disease, such as cancer (*Kozlovski et al., 2017*) and diabetes (*Dlamini et al., 2017*), but also occurs as a function of normal aging (*Heintz et al., 2017*; *Latorre and Harries, 2017*). Thus, a better understanding of the function and regulation of novel alternatively spliced IR isoforms may be useful in understanding the pathophysiology of insulin resistance.

Truncated IR isoforms that retain the ligand binding domain, but lack the intracellular signaling domain, have the potential to sequester insulin away from full length receptors thereby inhibiting signal transduction. For example, in *Drosophila*, an insulin binding protein (SDR) with significant homology to the extracellular domain of the fly insulin-like receptor (InR) has been shown to antagonize insulin signaling by binding insulin-like peptides (*Okamoto et al., 2013*). However, this variant is encoded by a distinct gene rather than arising via alternative splicing of the InR locus. In mammals,

intronic polyA activation has been proposed as a mechanism by which truncated tyrosine kinase receptors may be generated (*Vorlová et al., 2011*) and has been demonstrated to be functionally significant in the case of the mammalian platelet-derived growth factor receptor alpha (PDGFRα) receptor (*Mueller et al., 2016*). Although it is hypothesized that such a mechanism may exist to create truncated IR isoforms (*Vorlová et al., 2011*), their functional relevance has yet to be demonstrated.

In the nematode *C. elegans*, the *daf-2* gene encodes a tyrosine kinase receptor with sequence and structural homology to both the insulin and insulin-like growth factor-I (IGF-I) receptors in mammals (*Kimura et al., 1997*). Hypomorphic genetic mutations in *daf-2* lead to phenotypes such as developmental arrest and lifespan extension (*Gems et al., 1998*). *C. elegans* also has an unusually large number of insulin-like peptides in its genome which can act as both functional agonists and antagonists of insulin signaling (*Pierce et al., 2001*; *Li et al., 2003*; *Murphy et al., 2003*; *Murphy et al., 2007*; *Zheng et al., 2018*). Similar to other species, multiple isoforms of *daf-2* have been described, including *daf-2a* and *daf-2c* which are analogous to mammalian IR-A and IR-B (*Ohno et al., 2014*). Other isoforms of *daf-2* have also been suggested to exist (*Ohno et al., 2014*), including a truncated isoform, *daf-2b*, which was first identified based on expressed sequence tag (EST) cDNA evidence. However, when and where truncated IR isoforms, such as DAF-2B, are generated and what functional role truncated IR isoforms play remains unknown.

We reasoned that DAF-2B has the potential to act as a modifier of insulin signaling since it retains the ligand binding domain but lacks the signaling domain. We show a *daf-2b* transcript exists in *C. elegans* and *daf-2b* cDNA is detected at all larval stages, as well as in the alternative dauer larval stage. In vitro biochemical results indicated that DAF-2B dimerizes, suggesting that it could bind insulin-like peptides and thereby modulate insulin signaling by acting as a decoy receptor. Using a transgenic splicing reporter, we determined that *daf-2b* splicing in vivo was subject to temporal and spatial regulation, distinct from that of the *daf-2a* and *daf-2c* full-length receptor isoforms, most noticeably during the alternative dauer larval stage. Furthermore, characterization of an endogenous translational fusion protein confirmed that DAF-2B is expressed and likely acts as a secreted protein. Analysis of dauer formation and recovery, established paradigms for insulin-like peptide activity (*Cornils et al., 2011*), indicated that DAF-2B modifies insulin signaling both positively and negatively, in a manner consistent with the sequestration of insulin peptides. This mechanism was confirmed by co-expression of DAF-2B with both agonist and antagonist insulin-like peptides. Thus, our results indicate that a truncated IR isoform arising via alternative splicing represents a new fundamental principle for how the insulin signaling axis is regulated.

## Results

### *C. elegans* expresses a truncated isoform of *daf-2*

The *daf-2a* transcript spans 17 exons and encodes a 1846 amino acid (aa) protein (*Kimura et al., 1997*), while *daf-2c* includes an alternative exon, exon 11.5 (*Ohno et al., 2014*) (*Figure 1A*). *daf-2a* and *daf-2c* both encode an extracellular ligand binding domain (α subunit, exons 1–11/11.5) and a transmembrane and intracellular tyrosine kinase domain (β subunit, exons 12–17). We noted the existence of EST cDNAs (EC006316, EC004351) in Wormbase (www.wormbase.org) that aligned with *daf-2* exon 11, exon 11.5 and 128 bp of intronic sequence following exon 11.5 (*Figure 1—figure supplement 1*). Receptor tyrosine kinase EST sequences that span an intron/exon boundary as well as an upstream exon/exon boundary have been associated with truncated transcripts derived from activation of alternative intronic polyadenylation (polyA, AAUAAA) sites (*Vorlová et al., 2011*). Inspection of the *daf-2* genomic sequence indicates that the exon 11.5 5′ splice site (UUguaugga) diverges from the consensus (AGguaaguu) (*Zahler, 2012*) (*Figure 1B*). Failure to utilize this splice site would result in the addition of 46 bp of intronic sequence before reaching an in-frame stop codon (*Figure 1B*, *Figure 1—figure supplement 1*). In addition, downstream of this stop codon we observed multiple possible polyA sites, including a variant (AAUGAA) 84 bp away and matching sequence from the ESTs (*Figure 1—figure supplement 1*). These EST sequences previously formed the basis for the existence of the *daf-2b* isoform, which was predicted to share the first 11 exons with the *daf-2a* full-length receptor but lack the transmembrane and intracellular tyrosine kinase domains encoded by exons 12–17 (*Figure 1A and B*). Consequently, a *daf-2b* transcript would

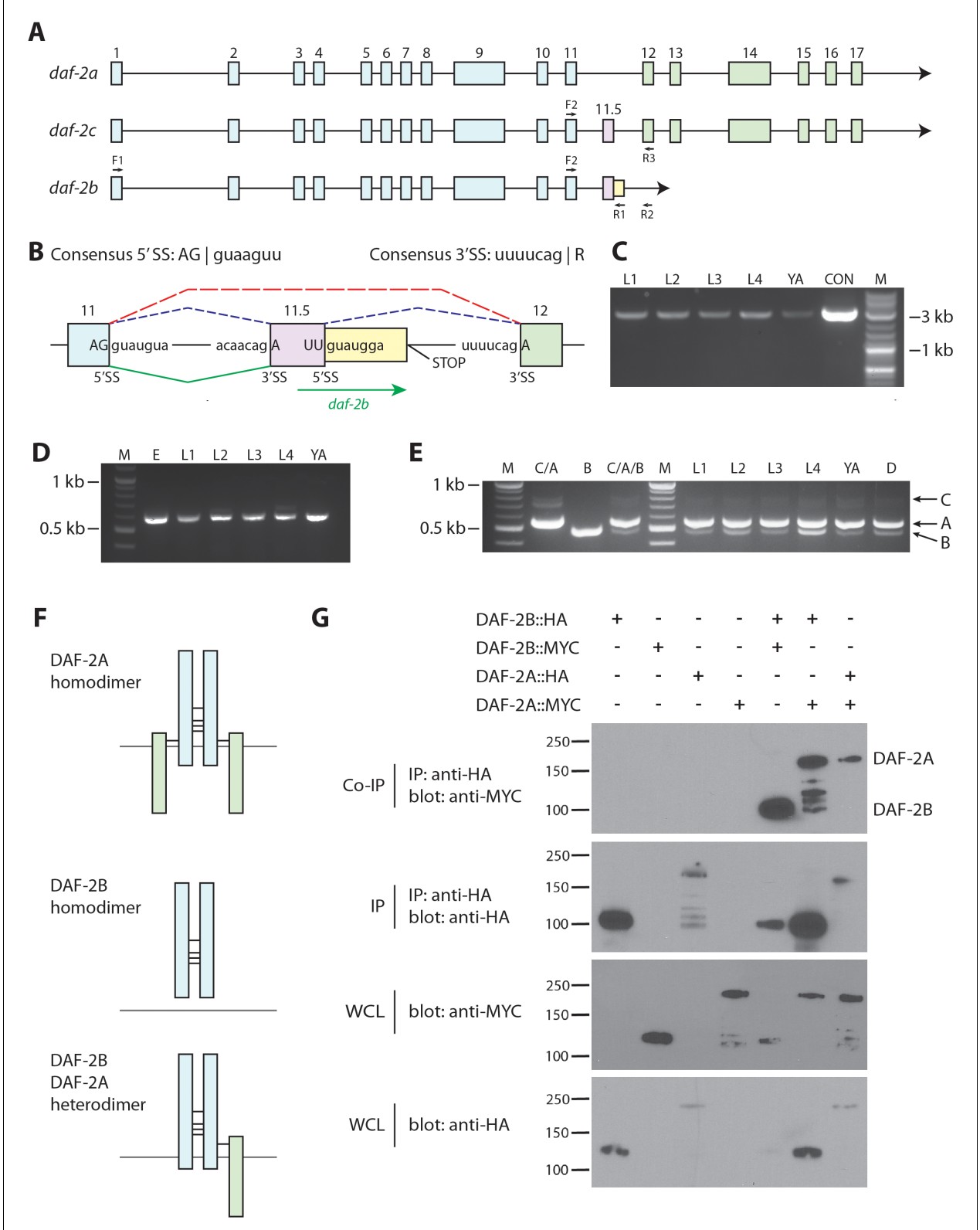

**Figure 1.** *daf-2b* encodes a truncated insulin receptor that is capable of dimerization. (**A**) Genomic organization of the *daf-2* locus. Exons shaded in blue encode the α subunit (extracellular domain) and those in green encode the β subunit (transmembrane and tyrosine kinase domains). The alternate cassette exon utilized in *daf-2c* (exon 11.5) is shown in pink. The *daf-2b* transcript is predicted to arise if splicing at the exon 11.5 5'SS is skipped leading to the addition of 46 bp of intronic sequence (shown in yellow) before an in-frame stop codon is reached. F1, F2, R1, R2 and R3 indicate the

*Figure 1 continued on next page*

*Figure 1 continued*

location of primers used in cDNA amplification. (B) Details of the *daf-2* genomic locus from exon 11 to exon 12. The sequence of 5' and 3' splice sites (SS) are indicated. Dotted lines indicate splicing events for *daf-2a* (red) and *daf-2c* (blue). Green solid lines indicate splicing events for generation of the *daf-2b* transcript. (C) PCR amplification of full-length *daf-2b* cDNA using primers F1 and R1 (panel A). M - molecular weight markers, L – larval stage, YA – young adults, CON – *daf-2b* cDNA from plasmid template. (D) PCR amplification of a *daf-2b* cDNA fragment encompassing exon 11 and the predicted 3' UTR using primers F2 and R2 (panel A). M - molecular weight markers, E – embryos, L – larval stage, YA – young adults. (E) Multiplex PCR of *daf-2a, daf-2b* and *daf-2c* from pooled cDNA (lanes marked C/A, B and C/A/B) and larval stages including dauer (D) using primers F2, R1 and R3 (panel A). (F) Schematic illustrating the possible formation of DAF-2A and DAF-2B homodimers and DAF-2A/DAF-2B heterodimers via formation of disulfide bonds at conserved cysteine residues. (G) Coimmunoprecipitation of epitope tagged DAF-2A and DAF-2B indicates the capacity to dimerize. Immunoprecipitates were subjected to SDS PAGE and blotted with anti-MYC (top panel) and anti-HA (second panel). Whole cell lysates (WCL) were blotted with anti-MYC and anti-HA (bottom two panels). Coimmunoprecipitation data are representative of 3 independent experiments.

The online version of this article includes the following figure supplement(s) for figure 1:

**Figure supplement 1.** Schematic of *daf-2b* cDNA.
**Figure supplement 2.** Ratio of *daf-2a* to *daf-2b*.

encode for a protein with 1020 aa that retains the complete extracellular ligand binding domain but would lack the intracellular signaling domain.

To determine if *daf-2b* is an expressed transcript in worms, we used RT-PCR to clone *daf-2b* cDNA fragments as well as the full length *daf-2b* cDNA from *C. elegans* polyadenylated RNA. To do so, we designed two reverse primers (R1, R2) specific to the predicted *daf-2b* transcript, which were used with two forward primers (F1, F2; **Figure 1A**). The first reverse primer (R1) was complementary to the retained intron sequence, and the second reverse primer (R2) was just upstream of the first predicted polyadenylation site (**Figure 1A**). We used an exon 1 forward primer (F1) and primer R1 to amplify full length *daf-2b* from total cDNA prepared from polyadenylated RNA isolated from synchronized, staged populations (**Figure 1C**). Sequencing confirmed that the 3 kb product was identical to the predicted full length *daf-2b* transcript. Using an exon 11-specific forward primer (F2) and primer R2, we also amplified a *daf-2b* cDNA fragment from all developmental stages that included the predicted 3' UTR (**Figure 1D**, **Figure 1—figure supplement 1**). To estimate the abundance of the *daf-2b* transcript relative to *daf-2a* and *daf-2c*, we performed multiplex PCR. We used a common exon 11 forward primer (F2) for all transcripts, and an exon 12 reverse primer (R3) to amplify *daf-2a* and *daf-2c*, and a *daf-2b*-specific reverse primer (R1). Comparison of the pairwise PCR reactions with the multiplex reaction using a pooled cDNA template indicated that the three amplicons could be differentiated (**Figure 1E**). Analysis of stage-specific cDNA showed that *daf-2b* is present at all life stages (**Figure 1E**) and *daf-2a* is between 4 and 7 times more abundant than *daf-2b* (**Figure 1—figure supplement 2**).

## DAF-2B forms homodimers as well as heterodimers with DAF-2A

In mammals, dimerization of the IR (via the formation of four disulfide bonds in the α subunit) is required for high affinity insulin binding (**Belfiore et al., 2009**). Since the cysteine residues mediating disulfide bonds are conserved in DAF-2B, we hypothesized that DAF-2B could also form homodimers as well as heterodimers with the full-length DAF-2A receptor (**Figure 1F**). To examine these possibilities, we performed coimmunoprecipitation experiments in HEK293T cells expressing epitope-tagged DAF-2A and DAF-2B (**Figure 1G**). When DAF-2B::HA was immunoprecipitated with an anti-HA antibody under non-reducing conditions, we observed robust coprecipitation of DAF-2B::MYC, indicating the presence of homodimers (**Figure 1G**). Similarly, when we immunoprecipitated DAF-2B::HA under non-reducing conditions coprecipitating DAF-2A::MYC was detected (**Figure 1G**). Consistent with previous biochemistry on mammalian IRs, we also detected DAF-2A::MYC coprecipitating with DAF-2A::HA (**Figure 1G**). These results indicate that DAF-2B can bind to both itself to form 2B/2B homodimers and bind to DAF-2A to form 2B/2A heterodimers.

## *daf-2b* splicing is subject to spatial and temporal regulation

To begin assessing the physiological role of DAF-2B, we examined where splicing that generates the *daf-2b* transcript may occur in vivo by constructing isoform-specific fluorescent splicing reporters (**Kuroyanagi et al., 2006**). To increase the accuracy and relevance of these *daf-2* splicing reporters, we used the native *daf-2* promoter to drive expression. To report on *daf-2a* and *daf-2c* expression,

we amplified a 3.2 kb *daf-2* genomic fragment comprised of exon 11 and exon 12, including the intervening intron, and placed it upstream of tdTomato and the *unc-54* 3'UTR (*Figure 2A*). This construct was capable of reporting splicing that would result in expression of both *daf-2a* and *daf-2c* but could not distinguish between each of these different splicing events. To examine splicing that generates *daf-2b*, we placed tdTomato between the retained intronic sequence at the end of exon 11.5 and the in-frame stop codon (*Figure 2B*). Thus, this reporter contains all the regulatory elements between exon 11 and exon 12 that likely influence *daf-2b* expression. In this construct, tdTomato expression will only be observed if splicing from exon 11 to 11.5 occurs and if the 5' splice site in exon 11.5 is skipped, leading to retention of intronic sequence that is specific to *daf-2b*

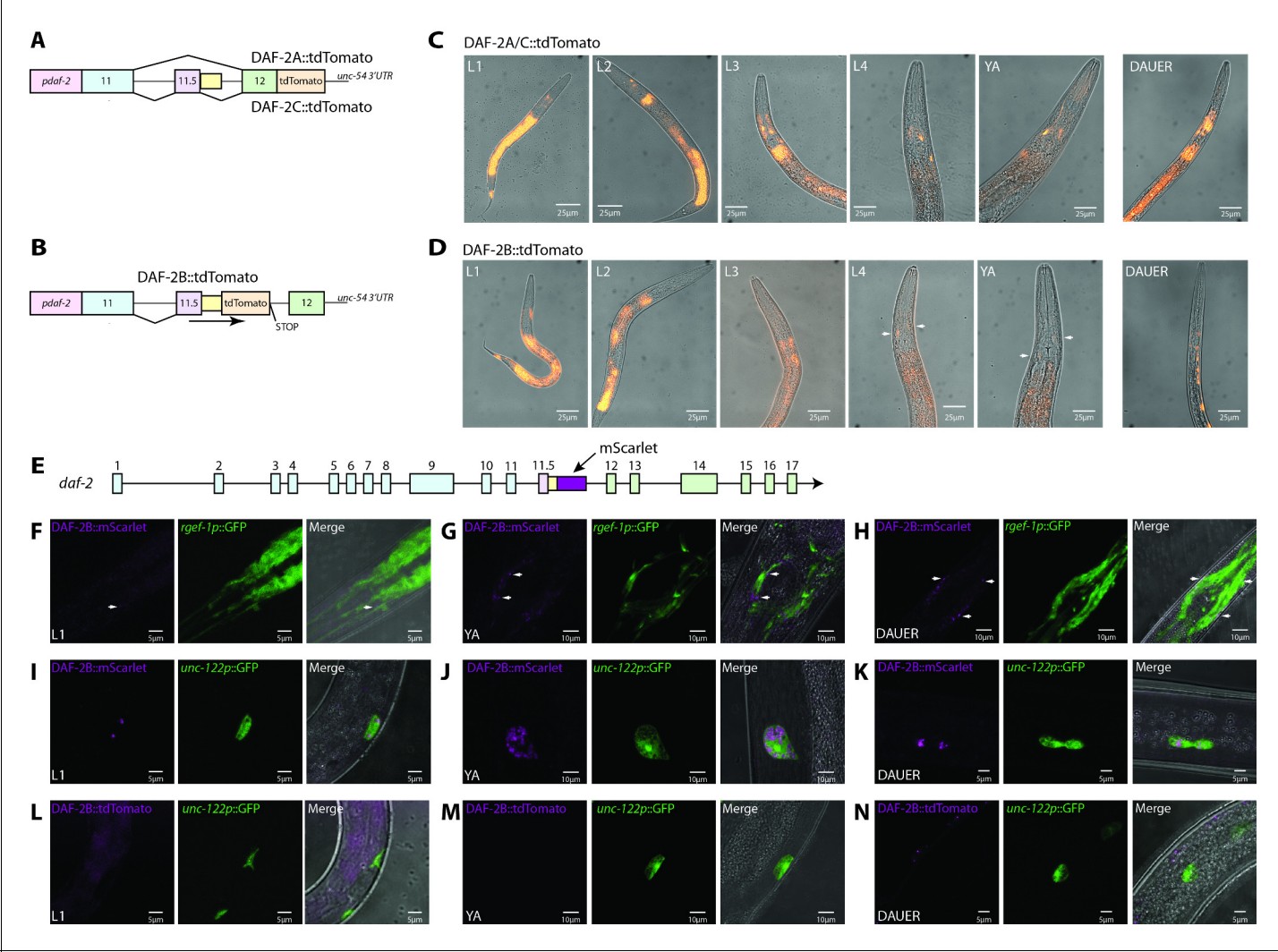

**Figure 2.** *daf-2b* splicing and expression are subject to temporal and spatial regulation. (**A**) Schematic illustrating the organization of the *daf-2a/c* splicing reporter. (**B**) Schematic illustrating the organization of the *daf-2b* splicing reporter. (**C**) Representative fluorescence images showing the expression of DAF-2A/C::tdTomato through reproductive development and in dauers in animals bearing extrachromosomal arrays. All images were taken with 1 s exposure. (**D**) Representative fluorescence images showing the expression of DAF-2B::tdTomato through reproductive development and in dauers in animals bearing an integrated array. All images were taken with 1.5 s exposure. (**E**) Schematic illustrating the location of mScarlet insertion into the *daf-2* genomic locus. Neuronal expression of DAF-2B::mScarlet in L1 (**F**), young adults (**G**) and dauers (**H**). Accumulation of DAF-2B::mScarlet in coelomocytes in L1 (**I**), young adults (**J**) and dauer (**K**) animals. DAF-2B::tdTomato from the transgenic splicing reporter is not expressed in coelomocytes in L1 (**L**), young adults (**M**) or dauers (**N**). Arrows indicate neuronal expression.

The online version of this article includes the following figure supplement(s) for figure 2:

**Figure supplement 1.** Tissue localization of the DAF-2A/C::tdTomato splicing reporter.

**Figure supplement 2.** Tissue localization of the DAF-2B::tdTomato splicing reporter.

(*Figure 2B*). We note that while this reporter cannot categorically prove that *daf-2b* splicing occurs in cells that exhibit tdTomato expression, prior studies in *C. elegans* showed this approach is valuable for studying alternative splicing (*Heintz et al., 2017*; *Kuroyanagi et al., 2006*; *Kuroyanagi et al., 2007*; *Kuroyanagi et al., 2010*).

In the presence of high food, low temperature and low population density, larval development in *C. elegans* proceeds through four stages (L1 to L4) before the final molt into a reproductive adult (*Byerly et al., 1976*). Under these conditions, the expression pattern of *daf-2a/c* from the transgenic splicing reporter matched that previously described for transcriptional *daf-2* reporters (*Fukuyama et al., 2015*; *Hunt-Newbury et al., 2007*) (*Figure 2C*, *Figure 2—figure supplement 1*). Like *daf-2a/c* splicing, the *daf-2b* splicing reporter was highly expressed in the hypodermis and intestine of L1 and L2 larvae (*Figure 2D*, *Figure 2—figure supplement 2*). However, the expression of the *daf-2b* transgenic splicing reporter declined across all larval stages into adulthood (*Figure 2D*).

Under adverse environmental conditions such as low food, high temperatures and high population density, worms adopt an alternate developmental pathway leading to formation of the dauer larva (*Cassada and Russell, 1975*). The dauer larva is non-feeding, non-reproducing, stress resistant and long-lived. In dauer larva carrying the *daf-2a/c* reporter, fluorescence was detected in neurons and in the intestine (*Figure 2C*, *Figure 2—figure supplement 1*). However, using the *daf-2b* splicing reporter in dauer larvae, we observed expression only in the nervous system (*Figure 2D*, *Figure 2—figure supplement 2*). These data indicate that splicing to generate a *daf-2b* transcript can occur in vivo in a tissue-specific and temporally regulated manner.

## DAF-2B is expressed in vivo and accumulates in coelomocytes

To establish whether DAF-2B protein is expressed in vivo, we used CRISPR-Cas9 editing to introduce the coding sequence for the mScarlet fluorescent protein into the *daf-2b* genomic locus immediately after exon 11.5 (*Figure 2E*). Expression of the DAF-2B::mScarlet translational fusion was faint in the nervous system of L1s, with stronger, punctate expression in young adults and dauers (*Figure 2F–H*). We did not see explicit colocalization of DAF-2B::mScarlet with the neuronal marker *rgef-1p:: GFP*, which could be due to the cytosolic location of GFP versus a cell surface localization of DAF-2B::mScarlet. However, the location of DAF-2B::mScarlet around the periphery of the GFP positive cells indicates that DAF-2B is localized in or around the nervous system. Expression of the endogenous translational fusion protein was not observed in the hypodermis or the intestine (data not shown). Interestingly, we did observe DAF-2B::mScarlet in the coelomocytes in L1s, young adults and dauers (*Figure 2I–K*), despite no expression of the *daf-2b* splicing reporter in these cells (*Figure 2L–N*). Coelomocytes are macrophage like cells that take up macromolecules from the pseudocoelomic space (*Fares and Greenwald, 2001*). Taken together, the robust, consistent accumulation of DAF-2B::mScarlet in coelomocytes and absence of *daf-2b* expression from the transgenic splicing reporter in these cells is evidence that secreted DAF-2B is likely to be phagocytosed by coelomocytes. Secreted DAF-2B accumulating in coelomocytes could come from any tissues or cells where our *daf-2b* splicing reporter is active. For example, this might explain why we observed expression of the *daf-2b* splicing reporter in the intestine, but couldn't detect DAF-2B::mScarlet in this tissue.

## DAF-2B influences dauer entry in temperature-sensitive insulin signaling mutants

Given the role of insulin signaling in the formation of the dauer larva, we hypothesized that the DAF-2B isoform could play an important role in modifying entry and/or exit from dauer. Reduced insulin signaling in hypomorphic *daf-2* IR mutants leads to temperature-sensitive dauer entry (*Gems et al., 1998*). We used transgenic extrachromosomal arrays to over-express DAF-2B in hypomorphic *daf-2* mutants and evaluated effects on dauer entry (*Figure 3*). In order to detect either increased or reduced dauer entry with DAF-2B overexpression, we used a semi-permissive temperature that yields an intermediate frequency of dauer entry in hypomorphic *daf-2* mutants. When a *daf-2b* cDNA was over-expressed using the *daf-2* promoter in this mutant background we observed increased dauer entry (*Figure 3A*). Expression of DAF-2B in the nervous system or the hypodermis also increased dauer entry, while expression in the intestine or the body wall muscles had no effect (*Figure 3A*).

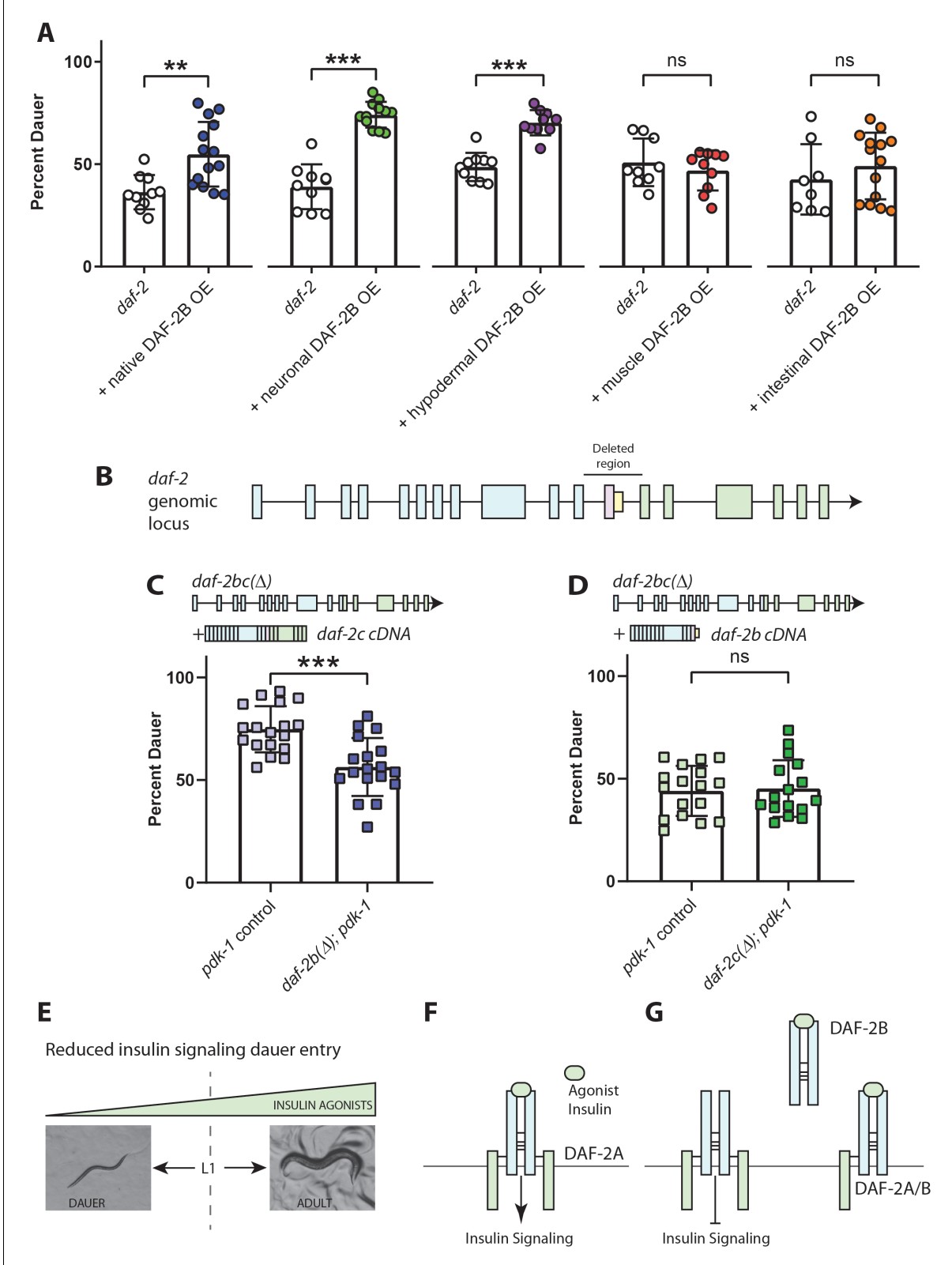

**Figure 3.** DAF-2B influences dauer entry in temperature-sensitive insulin-signaling mutants. (**A**) DAF-2B overexpression (OE) from the native *daf-2* promoter, in the nervous system or in the hypodermis enhances dauer formation in *daf-2* mutants at 23.2°C. There was no effect of DAF-2B expressed in the intestine or in the body wall muscle. Data are pooled from 2 (hypodermal promoter) or 3 (all other promoters) transgenic lines with at least four biological replicates per transgene. Students t-test ns – not significant, **p<0.01, ***p<0.001. Two additional trials with native, neuronal, muscle and

*Figure 3 continued on next page*

*Figure 3 continued*

intestinal DAF-2B OE showed similar effects. Raw data can be found in *Figure 3—source data 1*. (B) Schematic illustrating the region of the *daf-2* genomic locus deleted to generate *daf-2bc(Δ)*. (C) Genetic deletion of *daf-2b* suppresses dauer formation in *pdk-1* mutants at 26.8°C. Data are pooled from three independent trials with six biological replicates from one control and one *daf-2b(Δ)* strain per trial. Students t-test ***p<0.0001. Raw data can be found in *Figure 3—source data 2*. (D) Genetic deletion of *daf-2c* has no effect on dauer formation in *pdk-1* mutants at 26.8°C. Data are pooled from three independent trials with three biological replicates from each of two control and *daf-2c(Δ)* strains per trial. Students t-test ns – not significant. Raw data can be found in *Figure 3—source data 3*. (E) In temperature-sensitive hypomorphic insulin signaling mutants, the decision to enter dauer or develop into reproductive adults is dependent on the level of insulin agonism. (F) Under semi-permissive conditions, agonism at the insulin receptor promotes insulin signaling and reproductive growth. (G) DAF-2B, acting as a homodimer or a heterodimer, is predicted to sequester agonist insulin peptides, thereby inhibiting insulin signaling and promoting dauer formation. Error bars represent mean ± sd and data are shown as independent replicates from multiple transgenic lines.

The online version of this article includes the following source data for figure 3:

**Source data 1.** DAF-2B overexpression enhances dauer formation in *daf-2*(*e1368*).
**Source data 2.** *daf-2b* deletion suppresses dauer formation in *pdk-1*(*sa709*).
**Source data 3.** *daf-2c* deletion has no effect on dauer formation in *pdk-1*(*sa709*).

To test how the absence of DAF-2B affects dauer formation, we used CRISPR-Cas9 gene editing to generate a *daf-2b* deletion mutant. Given the splicing mechanism by which the DAF-2B isoform arises, it was not possible to generate a deletion of *daf-2b* without affecting the expression DAF-2C. Our approach therefore involved removing the entire intron between exons 11 and 12 in conjunction with homology directed repair to restore the *daf-2a* reading frame (*Figure 3B*). Since this deletion results in the loss of both the *daf-2b* and *daf-2c* isoforms, we also generated Mos single copy insertions (MosSCI) that express either *daf-2b* or *daf-2c* (*Figure 3C,D*). Using a combination of these reagents, we generated *daf-2b(Δ)* deletion animals in which *daf-2b* and *c* are deleted and MosSCI DAF-2C is expressed. Likewise, *daf-2c(Δ)* animals were generated by combining the *daf-2bc* deletion mutants with MosSCI DAF-2B expression. It was not possible to examine the effect of *daf-2b* deletion in the *daf-2* hypomorph background as both genetic lesions are extremely close together in the same gene. Therefore, to examine the effect of loss of *daf-2b* on dauer formation caused by hypomorphic insulin signaling, we crossed the deletion strains into a *pdk-1* mutant background. Previous work has shown that PDK-1 mediates DAF-2 insulin signaling during dauer formation (*Paradis et al., 1999*). There was a significant reduction in dauer entry in *daf-2b(Δ); pdk-1* animals compared with *pdk-1* control animals (*Figure 3C*). In contrast, there was no effect on dauer entry in *daf-2c(Δ); pdk-1* animals (*Figure 3D*).

The decision to proceed with reproductive growth or to enter dauer in temperature sensitive insulin-signaling mutants is driven primarily by the level of agonistic insulin peptides (*Cornils et al., 2011*) (*Figure 3E,F*). Our biochemical results indicated that DAF-2B can form homodimers (*Figure 1G*) and thus we suggest a model in which DAF-2B may sequester insulin peptides away from full length DAF-2 homodimeric receptors (*Figure 3G*). Alternatively, our observation that heterodimerization of DAF-2B occurs with full-length DAF-2A (*Figure 1G*) suggests DAF-2B could also dimerize with DAF-2A and act in a dominant negative manner (*Figure 3G*). Taken together, these observations indicate that under semi-permissive hypomorphic insulin signaling conditions DAF-2B can promote dauer entry by changing availability of agonist insulin peptides, thereby further reducing insulin signaling (*Figure 3G*).

## DAF-2B influences dauer recovery in temperature-sensitive insulin-signaling mutants

Next, we evaluated how DAF-2B affects recovery from dauer, a process that requires an increase in insulin signaling (*Figure 4A*) that is principally mediated by the agonistic insulin-like peptide *ins-6* (*Cornils et al., 2011*). Transgenic overexpression of DAF-2B from the native *daf-2* promoter was sufficient to inhibit dauer recovery in hypomorphic *daf-2* mutants (*Figure 4B*), as was expression in the nervous system, body wall muscle, hypodermis or intestine (*Figure 4C–F*). Conversely, dauer recovery was accelerated in *daf-2b(Δ); pdk-1* animals compared with their controls (*Figure 4G*), but not in *daf-2c(Δ); pdk-1* animals (*Figure 4H*). These results demonstrate that DAF-2B also has robust effects on dauer exit, that are consistent with DAF-2B restricting insulin signaling.

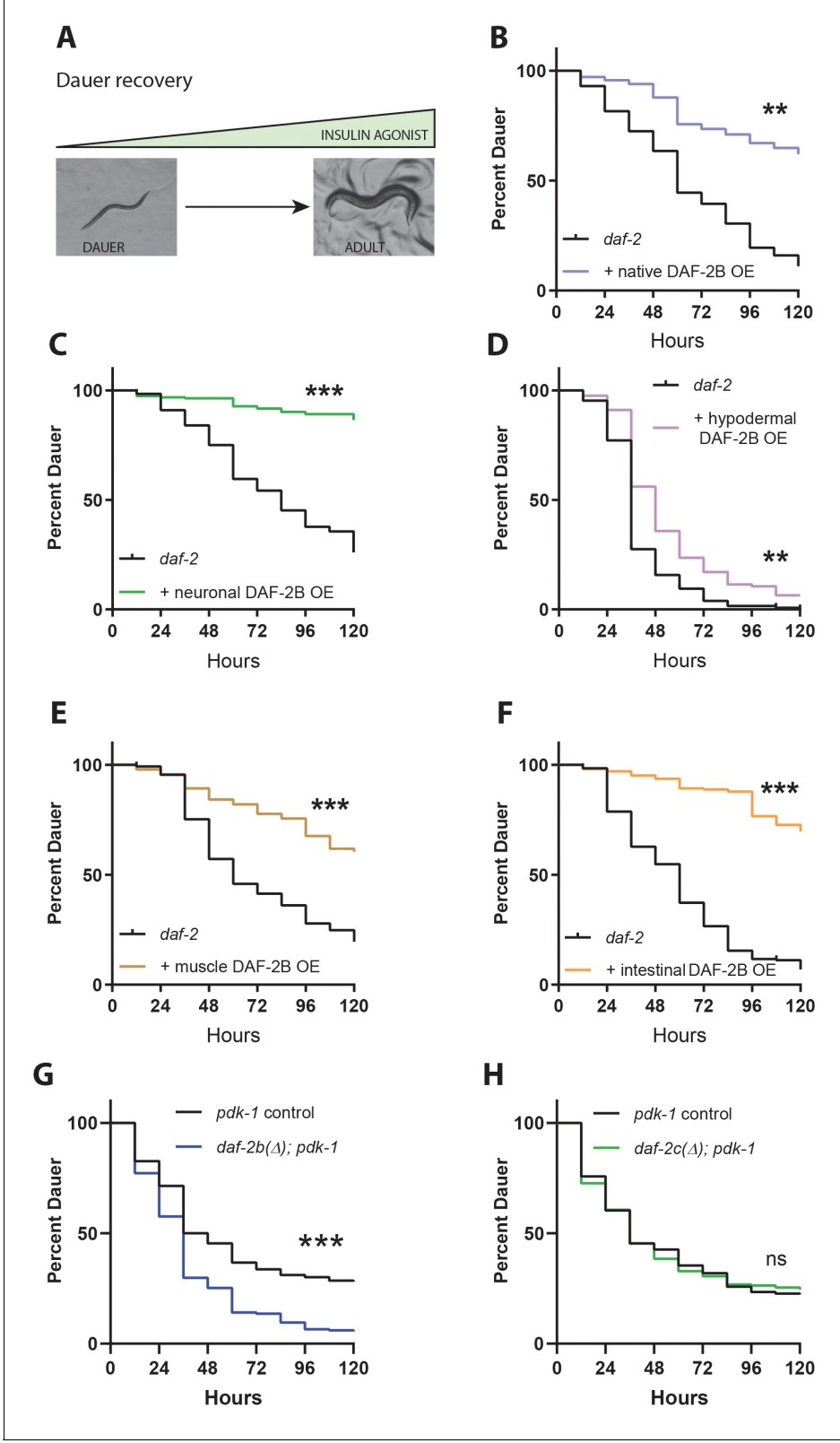

**Figure 4.** DAF-2B influences dauer recovery in temperature-sensitive insulin-signaling mutants. (A) Exit from the dauer larva into a reproductive adult requires an increase in the activity of insulin agonists. Overexpression of DAF-2B from the native *daf-2* promoter (B), in the nervous system (C), in the hypodermis (D), in muscle (E) or in the intestine (F) inhibits dauer recovery in *daf-2* mutants under permissive conditions. Data are pooled from three transgenic lines (except two for hypodermal and muscle promoters) with at least three technical replicates per transgene. Two additional trials with

*Figure 4 continued on next page*

*Figure 4 continued*

native, neuronal, muscle and intestinal DAF-2B OE showed similar effects. Raw data can be found in *Figure 4—source data 1*. (G) Genetic deletion of *daf-2b* promotes dauer exit in *pdk-1* mutants. Data are pooled from six technical replicates from one control and one *daf-2b(Δ)* strain. Two biological repeats showed similar effects. Raw data can be found in *Figure 4—source data 2*. (H) Genetic deletion of *daf-2c* has no effect on dauer exit in *pdk-1* mutants. Data are pooled from three technical replicates from each of 2 control and *daf-2c(Δ)* strains per trial. Two biological repeats showed similar effects. Log Rank test ns – not significant, **p<0.01, ***p<0.001. Raw data can be found in *Figure 4—source data 2*.

The online version of this article includes the following source data for figure 4:

**Source data 1.** DAF-2B overexpression inhibits dauer recovery in *daf-2(e1368)*.
**Source data 2.** *daf-2b* deletion enhances dauer recovery in *pdk-1(sa709)*.

## DAF-2B acts via sequestration of insulin peptides

We hypothesized that overexpressed DAF-2B sequesters agonist insulin peptides, thereby reducing insulin action and inhibiting insulin signaling (*Figure 5A*). If this is the case, excess agonist insulin peptides should overcome the effect of DAF-2B overexpression and restore insulin signaling (*Figure 5B*). *daf-28* encodes an agonist peptide that is required to proceed with reproductive growth and overexpression of DAF-28 has been shown to suppress dauer entry in response to elevated temperature (*Li et al., 2003*; *Cornils et al., 2011*). Consistent with this, we found that overexpression of DAF-28 suppressed the dauer constitutive phenotype of *daf-2* mutants at 23℃ (*Figure 5C*). As previously observed, overexpression of DAF-2B enhanced dauer entry in *daf-2* mutants, but this effect was suppressed when DAF-2B and DAF-28 were overexpressed together (*Figure 5C*). Another agonist insulin peptide, *ins-6*, has been shown to be involved in exit from the dauer stage (*Cornils et al., 2011*) and we observed that overexpression of INS-6 significantly accelerated dauer exit in *daf-2* mutants at 22.5℃ (*Figure 5D*), as expected. INS-6 overexpression was also capable of overcoming the inhibitory effects of DAF-2B on dauer recovery (*Figure 5D*). Taken together, these observations with DAF-28 and INS-6 overexpression indicate that excess insulin peptides can promote insulin signaling in the presence of DAF-2B.

In parallel, we used site-directed mutagenesis to engineer a point mutation in the L1 domain of DAF-2B that is thought to be important for insulin peptide binding. This point mutation was identified in the *daf-2(e979)* mutant (*Patel et al., 2008*) and mutations in this domain of the human IR are associated with severe insulin resistance (*Lahiry et al., 2010*). We therefore reasoned that transgenic animals expressing the mutant form of DAF-2B should have a reduced capacity to attenuate insulin signaling (*Figure 5E*). Consistent with this, we found that there was a significant reduction in the number of dauers formed when mutated DAF-2B was overexpressed in a *daf-2* background compared with wild type DAF-2B (*Figure 5F*).

## DAF-2B overexpression extends lifespan

Reduced insulin signaling is also associated with increased longevity in *C. elegans*. Therefore, we tested how increased DAF-2B affects lifespan. We found that wild-type animals overexpressing DAF-2B resulted in a robust increase in lifespan that was similar to *daf-2* mutants (*Figure 6A,B*, *Supplementary file 1*). While overexpression of DAF-2B using a pan-neuronal promoter showed the most prominent increase in lifespan (*Figure 6C*), smaller but significant increases in lifespan also occurred when DAF-2B was overexpressed using promoters for hypodermis, muscle and intestine (*Figure 6D–F*). This suggests that extra copies of DAF-2B can reduce insulin signaling by sequestering agonistic insulin peptides in adult animals to confer lifespan extension. In contrast, neither loss of *daf-2b* or *daf-2c* affected lifespan (*Figure 6G and H*, *Supplementary file 1*). Taken together with the dauer entry and exit results, these data provide compelling evidence that the DAF-2B isoform functions to modulate insulin signaling in response to insulin agonists.

## DAF-2B attenuates the activity of antagonistic insulin peptides

Although many of the *C. elegans* insulin peptides act as agonists of insulin signaling, several insulin peptides, including *ins-1* and *ins-18*, function as antagonists (*Pierce et al., 2001*; *Matsunaga et al., 2012*). In addition, it has been shown that a number of insulin peptides can act as both agonists and antagonists depending on the context (*Zheng et al., 2018*). We therefore explored whether DAF-2B

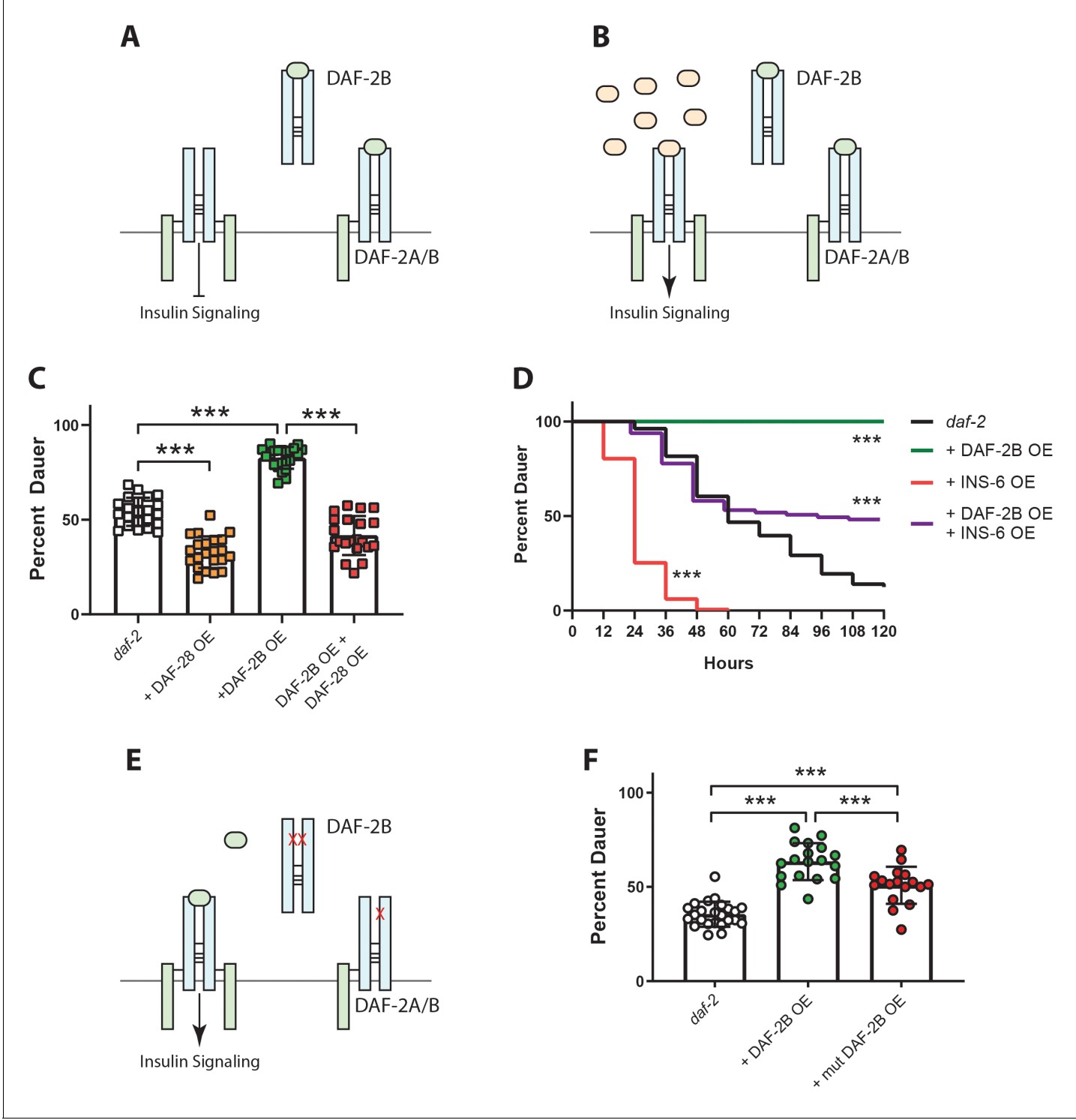

**Figure 5.** DAF-2B acts via sequestration of insulin peptides. (A) Model for the action of DAF-2B. Increased expression of DAF-2B leads to sequestration of insulin peptides away from full length receptors resulting in reduced insulin signaling. In this respect, DAF-2B may act as a homodimer or as a heterodimer with a full-length isoform. (B) A prediction of the model is that increased availability of insulin peptide agonists should restore insulin signaling. (C) Overexpression of the DAF-28 agonist peptide from the native *daf-28* promoter suppresses dauer formation in *daf-2* mutants, while overexpression of DAF-2B from an integrated neuronal *rab-3* promoter enhances dauer formation at 23°C. Combined overexpression of DAF-28 and DAF-2B reduces dauer formation compared with DAF-2B overexpression alone. Data are pooled from two independent trials with 12 biological replicates for *daf-2(e1368)* and the integrated neuronal DAF-2B OE line and two biological replicates from each of 5 or 6 DAF-28 OE extrachromosomal array lines. Post-hoc pairwise comparisons following one-way ANOVA ***p<0.0001. Raw data can be found in **Figure 5—source data 1**. (D)

*Figure 5 continued on next page*

*Figure 5 continued*

Overexpression of the INS-6 insulin peptide from the *rgef-1* promoter accelerates dauer recovery in *daf-2* mutants at 22.5°C. Combined overexpression of INS-6 and DAF-2B in *daf-2* mutants reverses the inhibitory effect of integrated neuronal DAF-2B expression on *daf-2* dauer recovery. Data are pooled from six technical replicates for *daf-2(e1368)* and the integrated neuronal DAF-2B OE line. For extrachromosomal array INS-6 OE lines, one replicate from each of 6 lines were pooled and at least three lines pooled for DAF-2B OE + INS-6 OE. Log Rank test – *daf-2 vs daf-2* + DAF-2B OE p<0.0001; *daf-2 vs daf-2* + INS-6 OE p<0.0001; *daf-2* + DAF-2B OE vs *daf-2* + DAF-2B OE + INS-6 OE p<0.0001. Two additional trials showed similar effects. Raw data can be found in *Figure 5—source data 2*. (E) Mutations in the ligand binding domain of DAF-2B that reduce affinity for insulin peptides should limit the ability of DAF-2B to inhibit insulin signaling. (F) Overexpression of wild type DAF-2B in the nervous system enhances dauer entry in *daf-2* mutants. The inhibitory effect of DAF-2B is attenuated by a point mutation that affects the L1 insulin binding domain. Data are pooled from two independent trials with six transgenic lines for DAF-2B::FLAG and DAF-2B(C196Y)::FLAG and 1–2 biological replicates per line. Post-hoc pairwise comparisons following one-way ANOVA ***p<0.0001. Error bars represent mean ± sd and data are shown as independent replicates from multiple transgenic lines. Raw data can be found in *Figure 5—source data 3*.

The online version of this article includes the following source data for figure 5:

**Source data 1.** Overexpression of the DAF-28 agonist insulin peptide suppresses the effects of DAF-2B OE in *daf-2*(*e1368*).
**Source data 2.** Overexpression of the INS-6 agonist insulin peptide suppresses the effects of DAF-2B OE in *daf-2*(*e1368*).
**Source data 3.** Overexpression of mutant DAF-2B in *daf-2*(*e1368*).

is also capable of attenuating the activity of antagonistic insulin peptides using the paradigm of pheromone-induced dauer formation.

In the natural environment, dauer formation is triggered by a reduction in food availability and an increase in the concentration of dauer pheromones (*Golden and Riddle, 1984*). Exposure to pheromone extracts leads to the down regulation of agonist insulin peptides, such as *daf-28,* and a relative increase in the activity of antagonistic peptides, such as *ins-1* and *ins-18* (*Pierce et al., 2001*; *Li et al., 2003*; *Matsunaga et al., 2012*) (*Figure 7A*). Thus, dauer entry in response to pheromone extracts involves a decrease in insulin signaling that is driven by a relative increase in activity of antagonistic insulin peptides (*Figure 7B*). Consistent with this, we found that transgenic overexpression of a *daf-2b* cDNA in wild type animals using the *daf-2* promoter reduced dauer entry in response to a pheromone extract (*Figure 7C*). Using tissue-specific promoters, expression of DAF-2B in the nervous system and the hypodermis, but not the intestine or the body wall muscles, was also sufficient to reduce dauer formation in response to pheromone extracts (*Figure 7C*). In opposition, genetic deletion of *daf-2b* enhanced dauer entry in the presence of pheromone extracts (*Figure 7D*), while loss of *daf-2c* had no effect on dauer entry (*Figure 7E*).

To test whether DAF-2B could sequester antagonist insulin peptides under these conditions, we generated INS-18 overexpressing animals and examined their ability to influence pheromone-induced dauer formation in the presence or absence of DAF-2B. When overexpressed alone, INS-18 enhanced dauer entry in the presence of pheromone, indicating antagonist activity (*Figure 7F*). In contrast, DAF-2B OE alone suppressed dauer formation as expected. However, INS-18 OE was unable to enhance pheromone-induced dauer formation in the presence of extra copies of DAF-2B (*Figure 7F*). These data suggest that under conditions in which insulin antagonists prevail, DAF-2B can also act as an insulin sensitizer by sequestering antagonistic insulin peptides.

## Discussion

We describe the cloning and functional characterization of DAF-2B, a truncated isoform of the *C. elegans* IR that arises via alternative splicing. The insulin signaling pathway is arguably the most well-studied pathway in the worm, yet our knowledge of the exact mechanisms by which insulin signals are transduced remains incomplete. The existence of a large number of insulin-like peptides and multiple signaling isoforms of the DAF-2 receptor suggests that precise control of receptor-ligand interactions is likely to be important in maintaining insulin specificity and facilitating differential IR signaling. Although this is achieved in part by spatial and temporal control of insulin peptide expression (*Ritter et al., 2013*; *Fernandes de Abreu et al., 2014*), our results now indicate that DAF-2B further modulates IR signaling, and provides an additional level of regulation in the worm insulin signaling pathway.

Our study focused on the hypothesis that DAF-2B might modulate insulin signaling by acting as a decoy receptor and sequestering insulin peptides away from full-length DAF-2 receptors. Indeed,

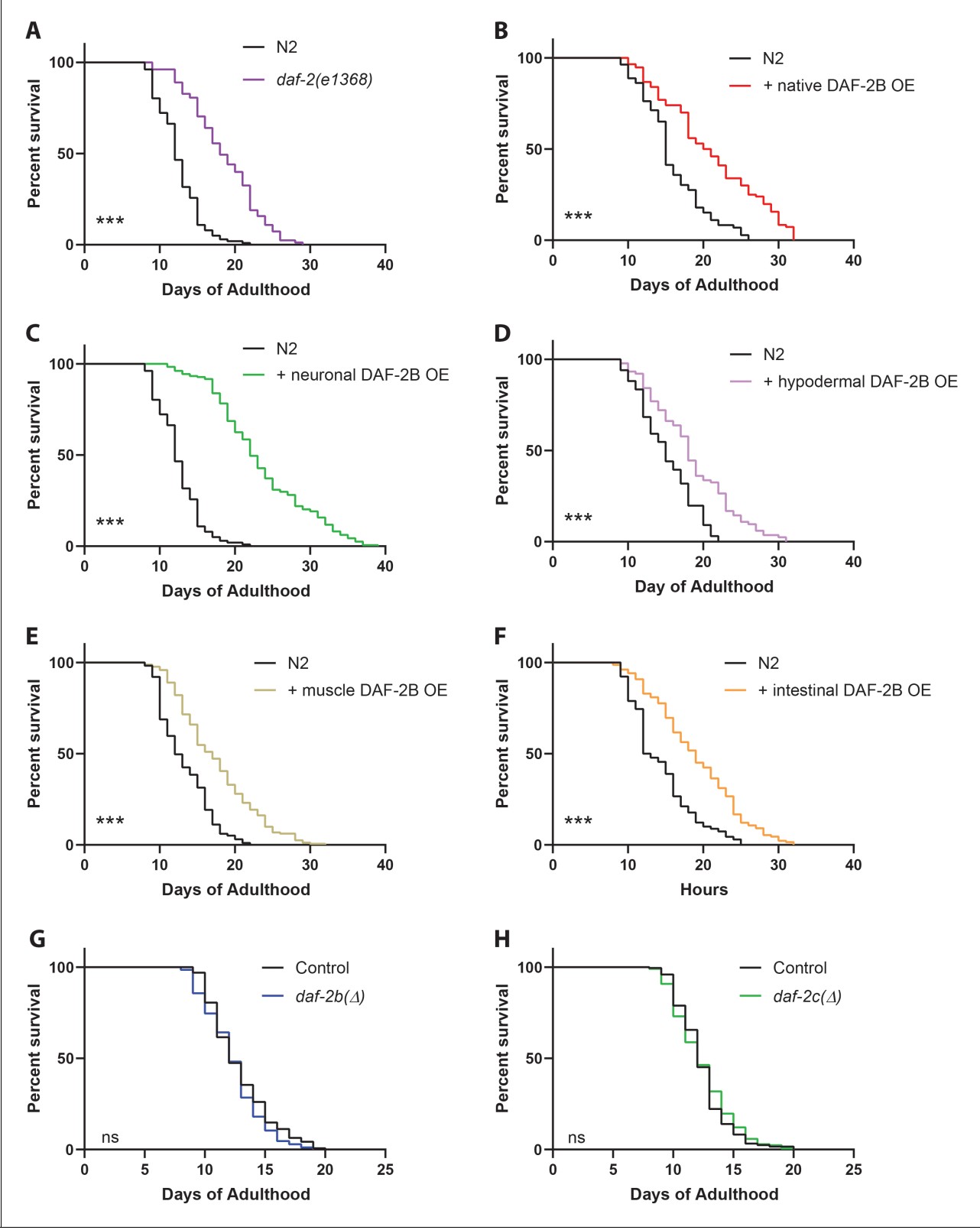

**Figure 6.** DAF-2B overexpression extends lifespan. (**A**) The *daf-2(e1368)* hypomorphic mutation confers lifespan extension. Overexpression of DAF-2B from the native *daf-2* promoter (**B**), in the nervous system (**C**), in the hypodermis (**D**), in the muscle (**E**) or in the intestine (**F**) extends lifespan in wild type N2 animals. Genetic deletion of *daf-2b* (**G**) and *daf-2c* (**H**) has no effect on lifespan in wild type N2 animals. Log Rank test ns – not significant,

*Figure 6 continued on next page*

*Figure 6 continued*

***p<0.001. Lifespan data are representative of n = 2 biological replicates for DAF-2B overexpressers and n = 3 for deletion strains. Summary data for all replicates are presented in **Supplementary file 1**.

prior work indicates that receptor tyrosine kinases in mammals, such as PDGFRα, deploy this strategy to restrict receptor signaling (**Mueller et al., 2016**). These truncated isoforms arise by intronic poly A activation (**Vorlová et al., 2011**) and the *daf-2b* EST sequence (EC004351) we identified in Wormbase spans exon 11, exon 11.5 and contains intronic sequence, suggestive of a similar mechanism. To monitor *daf-2b* expression in vivo, we first constructed a *daf-2b* splicing reporter, using an approach that has proven useful in understanding the differential regulation of other alternatively spliced transcripts in *C. elegans* (**Heintz et al., 2017**; **Kuroyanagi et al., 2006**; **Kuroyanagi et al., 2007**; **Kuroyanagi et al., 2010**). This revealed a tissue-specific expression pattern for *daf-2b* that both overlapped with and was distinct from *daf-2a* and *daf-2c* expression across life stages. This suggests *daf-2b* expression provides a mechanism to regulate insulin signaling activity in a tissue-specific and temporally-regulated manner.

To confirm endogenous expression of DAF-2B protein, we used CRISPR/Cas9 gene editing to insert mScarlet into the *daf-2* genomic locus such that fluorescence would only be observed when DAF-2B protein is generated. Interestingly, while some neuronal expression of DAF-2B::mScarlet was observed, consistent with the splicing reporter, we were unable to detect fluorescence in the hypodermis or intestine. However, DAF-2B::mScarlet was detected in coelomocytes in all larval stages, as well as in dauers and adults. Coelomocytes are macrophage like cells that are involved in uptake and degradation of macromolecules that are secreted into the pseudocoelomic space (**Fares and Greenwald, 2001**). Indeed, accumulation of GFP-tagged proteins in these cells has been used as a marker of neuropeptide secretion in *C. elegans*, including the insulin peptide DAF-28 (**Kao et al., 2007**; **Sieburth et al., 2007**). The lack of DAF-2B::tdTomato fluorescence from the splicing reporter in coelomocytes indicates that they are not a site of expression of DAF-2B, and thus, the appearance of DAF-2B::mScarlet in these cells supports the idea that DAF-2B may act as a secreted protein. However, it should also be noted that differences in the pattern of expression between the endogenous fusion protein and the splicing reporter could arise from the absence of important regulatory sequences and/or the presence of the *unc-54* 3' UTR in the splicing construct.

Coimmunoprecipitation from a cell-based expression system confirmed that DAF-2B can form homodimers, which is a prerequisite for high affinity insulin binding (**De Meyts and Whittaker, 2002**). Studies of the soluble ectodomain of the mammalian IR, which comprises the α subunit and the extracellular portion of the β subunit, have shown it binds insulin with high affinity, as do other truncated isoforms of the IR that lack the β-subunit (**Brandt et al., 2001**). Thus, DAF-2B dimers are likely to bind insulin peptides. Computational analysis (**Zhang, 2008**; **Eisenhaber et al., 2000**) indicates that the C-terminal extension of DAF-2B is unlikely to contain a transmembrane domain, or a GPI anchoring motif. Therefore, DAF-2B homodimers could act as secreted, soluble decoy receptors. This model is well supported by the accumulation of DAF-2B::mScarlet in coelomocytes. However, cell-based experiments also demonstrated that DAF-2B can form heterodimers with full length DAF-2A. This might act as a second mechanism to restrict insulin signaling, as DAF-2B/2A heterodimeric receptors would be unlikely to transduce signals but should bind insulin peptides. More work will be required to elucidate the precise details of DAF-2 molecular species in vivo and their physiological relevance.

We used a combination of overexpression and genetic deletion to demonstrate a physiological role for the DAF-2B isoform. While expression of DAF-2B generally declined with reproductive growth, expression was retained in the dauer larva, specifically in neurons, suggesting a physiological role in regulating this process. Ohno and colleagues previously found that expression of *daf-2b* cDNA was unable to rescue the dauer phenotype of a hypomorphic *daf-2(e1370)* mutant (**Ohno et al., 2014**). This is not surprising, as DAF-2B lacks an intracellular signaling domain. We took a very different approach and focused on examining whether DAF-2B modulates insulin signaling. We found that overexpression of DAF-2B promoted dauer entry and inhibited dauer recovery in a temperature sensitive *daf-2* hypomorph with reduced insulin signaling. These findings are consistent with DAF-2B further reducing insulin signaling. We also examined phenotypes of a *daf-2b* loss of function mutant generated using CRISPR/Cas9 gene editing (**Paix et al., 2015**; **Paix et al., 2014**;

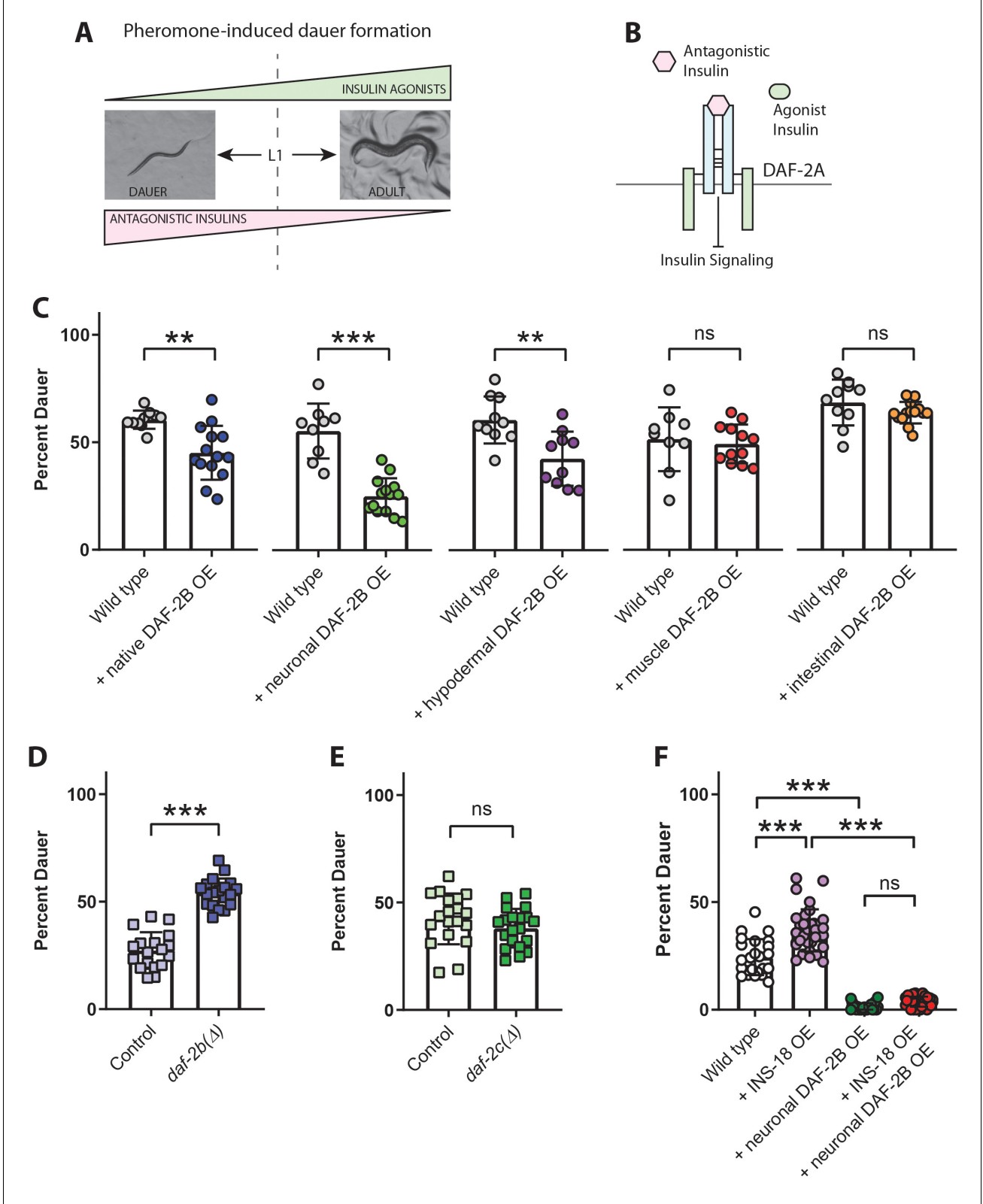

**Figure 7.** DAF-2B attenuates the activity of antagonistic insulin peptides. (**A**) In the presence of the dauer-inducing pheromone extracts, the decision in the first larval stage (L1) to proceed with reproductive growth or enter the dauer stage is dependent on the balance between antagonistic and agonistic insulin peptides. (**B**) Increased activity of antagonistic insulin peptides inhibits insulin signaling to promote dauer entry. (**C**) DAF-2B overexpression (OE) from the native *daf-2* promoter, in the nervous system, or in the hypodermis suppresses pheromone-induced dauer formation in wild type animals.
*Figure 7 continued on next page*

*Figure 7 continued*

There was no effect of DAF-2B expressed in the intestine or in the body wall muscle. Data are pooled from 2 (hypodermal promoter) or 3 (all other promoters) transgenic lines with at least four biological replicates per transgene. Students t-test ns – not significant, **p<0.01, ***p<0.001. Similar effects were observed in an additional trial for muscle, intestine and neuronal promoters and 1/2 additional trials for the native promoter. Raw data can be found in *Figure 7—source data 1*. (D) Genetic deletion of *daf-2b* enhances dauer formation in the presence of pheromone extracts. Data are pooled from two independent trials with three biological replicates from each of two control and two *daf-2b(Δ)* strains per trial. Students t-test ***p<0.001. Raw data can be found in *Figure 7—source data 2*. (E) Genetic deletion of *daf-2c* has no effect on dauer formation in the presence of pheromone extracts. Student t-test ns – not significant. Raw data can be found in *Figure 7—source data 3*. (F) Overexpression of the INS-18 antagonist insulin peptide from the native *ins-18* promoter enhances pheromone-induced dauer formation in wild type animals, while overexpression of DAF-2B from an integrated neuronal *rab-3* promoter suppresses dauer formation. Combined overexpression of INS-18 and DAF-2B also suppresses dauer formation compared with INS-18 overexpression alone. Data are pooled from two independent trials with 12 biological replicates for wild type, 11 or 12 biological replicates for the integrated neuronal DAF-2B OE line and 2 or three biological replicates from each of 6 INS-18 OE extrachromosomal array lines. Post-hoc pairwise comparisons following one-way ANOVA ***p<0.0001. Raw data can be found in *Figure 7—source data 4*. Error bars represent mean ± sd, and data are shown as independent replicates from multiple transgenic lines.

The online version of this article includes the following source data for figure 7:

**Source data 1.** DAF-2B overexpression suppresses pheromone-induced dauer formation in wild type.

**Source data 2.** *daf-2b* deletion enhances pheromone-induced dauer formation in wild type.

**Source data 3.** *daf-2c* deletion has no effect on pheromone-induced dauer formation in wild type.

**Source data 4.** DAF-2B overexpression suppresses the effect of INS-18 OE in pheromone-induced dauer formation in wild type.

*Ward, 2015*). Importantly, *daf-2b* loss of function resulted in the opposite phenotype to DAF-2B overexpression studies. Loss of *daf-2b* in the *pdk-1* insulin-signaling mutant resulted in suppression of dauer formation and promoted recovery from dauer. These latter data are consistent with a model in which loss of *daf-2b* promotes the activity of agonist insulin peptides.

In order to test the insulin sequestration model, we reasoned that if DAF-2B binds endogenous insulins to prevent signal transduction, then overexpression of an agonist insulin peptide should restore signaling. We were able to demonstrate this in a *daf-2* hypomorph using DAF-28 in the context of dauer entry and using INS-6 in a dauer recovery paradigm. In these experiments we used an integrated transgenic line expressing DAF-2B in the nervous system under the control of the *rab-3* promoter, while DAF-28 and INS-6 were expressed from extrachromosomal arrays using the *daf-28* and *rgef-1* promoters respectively. This approach mitigates the possibility that the reduced DAF-2B effect is a function of promoter dilution and suggests that DAF-2B is indeed sequestering insulin peptides. We also generated transgenic lines expressing a mutant form of DAF-2B that is predicted to compromise insulin binding. Dauer formation was significantly reduced in transgenic lines expressing the mutant DAF-2B compared with lines overexpressing the wild type DAF-2B, supporting the idea that DAF-2B mediates its effects through insulin sequestration. However, the fact that mutant DAF-2B still significantly enhanced dauer formation compared with *daf-2* controls suggests that there may also be insulin-independent effects of DAF-2B, such as via heterodimerization with full length receptors. This latter mechanism would be consistent with our observation of DAF-2A/DAF-2B heterodimer formation in coimmunoprecipitation studies. Furthermore, this mechanism may explain the appearance of punctate DAF-2B::mScarlet observed in neuronal tissues, which could be membrane bound DAF-2B in the form of a heterodimer with a full length receptor isoform.

In contrast to the temperature sensitive insulin signaling mutants, we observed that DAF-2B overexpression suppressed dauer formation in the pheromone paradigm, suggesting that it can also promote insulin signaling. This is further supported by the observation that loss of *daf-2b* enhanced dauer entry in response to pheromone. At first glance this seems paradoxical, until placed in the context of what is known about insulin peptide activity under these conditions. Upon exposure to dauer-inducing pheromone extracts, expression of agonist insulins, such as *daf-28* and *ins-6*, is reduced (*Li et al., 2003*; *Cornils et al., 2011*). In contrast, expression of antagonist insulins, such as *ins-1* and *ins-18*, remains stable (*Pierce et al., 2001*; *Cornils et al., 2011*) or is increased (*Matsunaga et al., 2012*), with the net result that insulin antagonism prevails and signaling through DAF-2 is diminished. However, in a *daf-2* mutant background loss of *ins-1* has only a modest effect on dauer entry at semi-permissive temperatures, while loss of the agonist peptide *daf-28* has a strong enhancer effect on dauer formation (*Cornils et al., 2011*). This means that temperature-dependent dauer formation in insulin signaling mutants is a consequence of reduced insulin

agonism, rather than a relative increase in antagonist activity. Thus, our results suggest that loss of DAF-2B in these hypomorphic insulin signaling mutant backgrounds suppresses dauer formation due to increased agonist activity, while extra copies of DAF-2B enhances dauer entry through agonist sequestration (*Figure 8A*). In contrast, in wild type animals exposed to pheromone, loss of DAF-2B enhances dauer entry due to increased activity of insulin peptide antagonists such as INS-18 (*Figure 8B*). On the other hand, increased DAF-2B suppresses pheromone-induced dauer formation via sequestration of insulin antagonists (*Figure 8B*).

Overexpression studies have indicated that a number of insulin peptides in *C. elegans* can have mixed function, in that they act as agonists or antagonists depending on the context (*Zheng et al., 2018*). This functional or phenotypic antagonism suggests that DAF-2B should be equally capable of sequestering antagonist peptides, as well as agonist peptides. This is supported by our observation that DAF-2B overexpression reduced dauer formation in wild type animals exposed to dauer-inducing pheromone. Furthermore, when we over-expressed INS-18, an antagonist peptide that enhances pheromone-induced dauer formation, we found that DAF-2B was still able to suppress dauer entry. One interpretation of this is that the binding capacity of DAF-2B in this transgenic line is sufficient to sequester the additional INS-18 arising from overexpression. Another possibility is that sequestration of INS-18 also affects the availability of agonist peptides. In this respect, it is noteworthy that the expression of INS-18 is reciprocal to that of the agonist INS-7 (*Murphy et al., 2007*; *Shaw et al., 2007*). Thus, reduced INS-18 signaling due to DAF-2B sequestration could lead to an upregulation of INS-7 expression, thereby further suppressing dauer formation.

Using tissue specific promoters, we found that neuronal and hypodermal expression of DAF-2B, but not intestinal or muscle, was sufficient to influence dauer entry. Thus, although DAF-2B::mScarlet CRISPR experiments support a model in which DAF-2B is secreted, we think that DAF-2B is acting locally rather than systemically, with outcomes depending on the prevailing insulin milieu. For example, neuronal changes in insulin peptide expression and secretion are known to be an important driver of the decision to proceed with reproductive growth or entry the dauer stage (*Murphy and Hu, 2013*). Thus, the neuronal effect of DAF-2B on dauer entry is not unexpected, and the hypodermal effect is perhaps due to the close proximity of these two tissues. However, it is also possible that the failure of the intestinal and muscle drivers to elicit a dauer formation phenotype is an artefact of how these heterologous promoters function during dauer entry. On the other hand, expression of DAF-2B from any tissue was sufficient to influence dauer recovery and lifespan. Hypomorphic *daf-2* mutations that lead to reduced insulin signaling also confer lifespan extension in *C. elegans* (*Kimura et al., 1997*; *Patel et al., 2008*; *Kenyon et al., 1993*). In these animals *ins-7* gene expression is repressed and *ins-7* RNAi is sufficient to extend lifespan (*Murphy et al., 2003*). Moreover, *ins-7* is not only expressed in the nervous system but also in the intestine and has been proposed to form part of a feed-forward loop that amplifies or abrogates insulin signaling (*Murphy et al., 2007*). Thus, the robust effect of DAF-2B expression from multiple tissues on longevity could be mediated via sequestration of agonists such as INS-7. Interestingly, loss of *daf-2b* did not have an appreciable effect on lifespan. Although DAF-2B::mScarlet was observed in coelomocytes of adult animals, the level of expression of the splicing reporter was low in these animals. Thus, under physiological conditions endogenous DAF-2B may be more important for regulating dauer formation than lifespan.

The discovery that the truncated DAF-2B isoform in *C. elegans* undergoes changes in expression during dauer formation and functions to regulate entry and exit from this alternative life-stage indicates DAF-2B is an important modulator of insulin signaling. Given evidence for conserved, truncated IR isoforms in other systems (*Västermark et al., 2013*; *Vorlová et al., 2011*; *Okamoto et al., 2013*), our findings have potentially revealed a fundamental principle for how insulin signaling is regulated. Furthermore, while the use of alternative intronic polyadenylation sites has been shown to generate shortened transcripts for receptor tyrosine kinases (*Vorlová et al., 2011*; *Lejeune and Maquat, 2005*), our study now provides the first evidence that this occurs in vivo to affect IRs. Further study of how alternative splicing affects IR signaling in other systems is now warranted and may uncover regulatory mechanisms similar to DAF-2B that could play an important role in normal physiology and pathophysiology of metabolic disease.

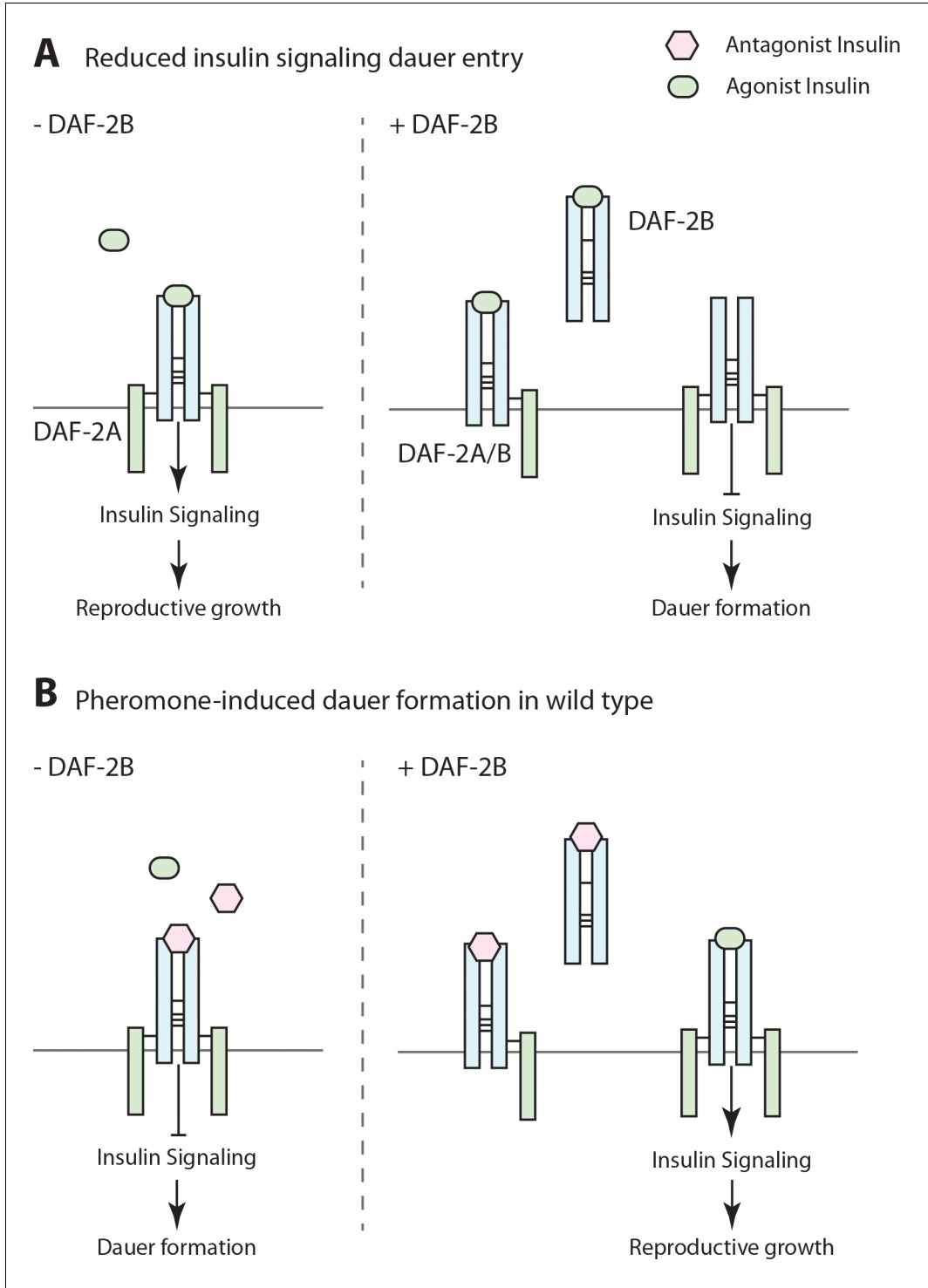

**Figure 8.** Model for the effect of DAF-2B on dauer formation. (**A**) In temperature-sensitive hypomorphic mutants that affect the insulin signaling pathway, such as *daf-2(e1368)* and *pdk-1(sa709)*, dauer entry is driven by a reduction in agonist insulin peptides. Under these conditions, loss of *daf-2b* promotes reproductive growth by enhancing insulin sensitivity, while extra-copies of DAF-2B enhance dauer entry by sequestering agonist insulins. (**B**) Pheromone-induced dauer formation in wild type worms is driven by an increase in activity of antagonist insulin peptides. In this paradigm, loss of *daf-2b* enhances dauer formation. Conversely, overexpression of DAF-2B promotes reproductive growth via sequestration of insulin antagonists.

# Materials and methods

## Key resources table

| Reagent type (species) or resource | Designation | Source or reference | Identifiers | Additional information |
|---|---|---|---|---|
| Gene (*Caenorhabditis elegans*) | *daf-2* | www.wormbase.org | WBGene00000898 | |
| Strain, strain background (*Escherichia coli*)) | OP50 | *Caenorhabditis* Genetics Center (CGC) | OP50 | |
| Strain, strain background (*C. elegans*) | *C. elegans* strains used and generated in this study | *Caenorhabditis* Genetics Center (CGC) and this paper | | *Supplementary file 4* |
| Cell line (human) | HEK293T/17 | ATCC | CRL-11268 | This cell line has been re-verified and has tested negative for mycoplasma |
| Antibody | Rabbit polyclonal anti-HA | Life Technologies | 715500 | IP: 3 ul for 20 mg protein |
| Antibody | Mouse monoclonal anti-myc | Cell Signaling Technology | 2276S | Dilution 1:1000 |
| Antibody | Mouse monoclonal anti-HA | Cell Signaling Technology | 2367S | Dilution 1:1000 |
| Antibody | Goat HRP-conjugated anti-mouse IgG | Fisher Scientific | Cat #9491974 | Dilution 1:4000 |
| Peptide, recombinant protein | Protein G agarose | Sigma | 11719416001 | |
| Recombinant DNA reagent | Plasmids generated in this study | This paper | | *Supplementary file 3* |
| Recombinant DNA reagent | Primers used in this study | This paper | | *Supplementary file 2* |
| Software, algorithm | Snapgene | GSL Biotech | RRID:SCR_015052 | https://www.snapgene.com/ |
| Software, algorithm | Prism 8 | Graphpad Software | RRID:SCR_002798 | https://www.graphpad.com/scientificsoftware/prism/ |
| Software, algorithm | Illustrator CS5 | Adobe | RRID:SCR_010279 | https://www.adobe.com/products/illustrator.html |
| Software, algorithm | Photoshop CS5 | Adobe | RRID:SCR_014199 | https://www.adobe.com/products/photoshop.html |
| Software, algorithm | Zen Digital Imaging for Light Microscopy | Zeiss | RRID:SCR_013672 | http://www.zeiss.com/microscopy/en_us/products/microscope-software/zen.html#introduction |
| Software, algorithm | ImageJ | ImageJ | RRID:SCR_003070 | https://imagej.net/ |

## Plasmids

### *daf-2a/c*::tdTomato splicing reporter

3 kb of the *daf-2* promoter, amplified using primers that introduced a 5' HindIII site and a 3' AgeI (primers 1 and 2), was cloned into pBGY487. A *daf-2* genomic fragment corresponding to exon 11 and exon 12, including the intervening intronic sequence with AgeI/KpnI overhangs and an ATG translation site was generated by PCR (primers 3 and 4). This fragment plus a KpnI-TdTomato::*unc-54* 3'UTR-SpeI fragment (primers 5 and 6) were cloned into pBGY487 *daf-2p* cut with AgeI/SpeI in a 3-piece ligation to generate pMGL83 *daf-2p::daf-2a/c::tdTomato* minigene.

### *daf-2b*::tdTomato splicing reporter

A *daf-2* fragment corresponding to exon 11-intron-exon 11.5 was amplified using primers that introduced a 5' AgeI site with an ATG translation start site and 3' KpnI site (primers 7 and 8) and cloned into pBGY487 *daf-2p*. tdTomato was amplified from pKBI using primers that introduced a 5' KpnI site and a 3' SalI site (primers 9 and 10) and cloned into pBGY487 *daf-2p* (3 kb)::exon11-intron-

exon11.5. A *daf-2* fragment consisting of exon 12 with the preceding intron plus the *unc-54* 3'UTR (primers 11 and 12) was then cloned into pBGY487 *daf-2p* (3 kb)::exon11-intron-exon11.5::tdTomato to generate pMGL86 *daf-2p::daf-2b*::tdTomato minigene.

## DAF-2B overexpression vectors

We obtained a full length *daf-2a* cDNA upstream of the *unc-54* 3'UTR in the pTG54 plasmid from Rene Garcia at Texas A & M University. To generate a *daf-2b* cDNA we synthesized a 600 bp construct that consisted of the last 51 nucleotides of exon 11, which contains an AjuI restriction site, the 292 bp exon 11.5 sequence, including the stop codon, and 257 bp of the *unc-54* 3'UTR, which contains an NheI restriction site. pTG54 was then digested with AjuI and NheI to remove exons 12–17 plus part of the *unc-54* 3'UTR and the new AjuI/NheI fragment containing exon 11.5 and the stop codon was subcloned. The resulting *daf-2b* plasmid was sequenced to confirm the presence of *daf-2b* without mutations.

For tissue-specific overexpressors we first generated a new destination vector with optimized restriction sites. GFP was PCR amplified from pPD95.75 using primers that introduced an SbfI – AgeI- NotI polylinker at the 5' end and an NheI site at the 3' end (primers 13 and 14) and cloned into pPD49.26 cut with SbfI and NheI. Tissue-specific promoters driving GFP were PCR amplified (*daf-2p* – primers 1 and 2; *rab-3p* – primers 15 and 16; *myo-3p* – primers 17 and 18; *ges-1p* – primers 19 and 20; *tag-335p* – primers 21 and 22, *rgef-1p* – primers 23 and 24; *unc-122p* – primers 25 and 26; *dpy-7p* – primers 27 and 28) and cloned into pPD49.26 AgeI NotI GFP stop with the appropriate restriction enzymes. *daf-2b* cDNA was amplified using primers 29 and 30 and cloned into pPD49.26 cut with SbfI and NheI. A NotI-*daf-2b*-KpnI fragment was cloned into tissue-specific expression vectors to generate *daf-2p::daf-2b* (native), *rab-3p::daf-2b* (neuronal), *myo-3p::daf-2b* (muscle), *ges-1p::daf-2b* (intestinal) and *tag-335p::daf-2b* (hypodermal).

*daf-2b* cDNA with a C-terminal FLAG tag was generated using primer pairs 31–32 and 33–34 with the PCR products cloned into pPD49.26 *rab-3p*. Mutant DAF-2B was generated by introducing the e979 mutation (C196Y) by site-directed mutagenesis using primers 35 and 36.

## Insulin peptide overexpressing vectors

The *daf-28* promoter was amplified with SbfI/AgeI overhangs (primers 37–38) and genomic *daf-28* was amplified with AgeI/KpnI overhangs (primers 39–40) before cloning into pPD49.26 AgeI NotI GFP stop. Genomic *ins-6* was amplified with AgeI and KpnI overhangs (primers 41–42) and cloned into pPD49.26 *rgef-1p*::GFP in place of the GFP sequence. The *ins-18* promoter plus genomic *ins-18* was amplified as a single fragment with SbfI and KpnI overhangs (primers 43–44) and cloned into pPD49.26 AgeI NotI GFP stop.

## Immunoprecipitation vectors

To express *daf-2b* and *daf-2a* in HEK cells for immunoprecipitation studies, we generated plasmids expressing each protein with C-terminal HA and Myc tags. pCMV HA (a gift from Roy Smith) was cut with ApaI and EcoR1, blunted and religated to remove N-terminal HA. Q5 site directed mutagenesis was used to generate pCMV HA-C (primers 45 and 46) and pCMV Myc-C (primers 47 and 48). *daf-2b* without a stop codon was amplified with 5'BglII and 3'NotI sites (primers 49 and 50) and cloned into pCMV HA-C. After sequencing, *daf-2b* was cloned into pCMV Myc-C using BglII and NotI. An AjuI/NotI fragment from *daf-2a* corresponding to exon 11 to exon 17 was cloned into pCMV *daf-2b*::HA-C and pCMV *daf-2b*::Myc-C to generate pCMV *daf-2a*::HA-C and pCMV *daf-2a*::Myc-C.

## *daf-2b* and *daf-2c* vectors for MosSCI

The *daf-2c* rescue plasmid, pPD49.26 *daf-2p::daf-2c* cDNA, was generated by NEB HiFi DNA assembly with primers designed using NEBuilder. *daf-2* exon 1–11.5 was amplified from *daf-2b* cDNA using primers that introduced a 5'overhang complementary to *daf-2p* and a 3' overhang complementary to *daf-2* exon 12 (Fragment 1, primers 51 and 52). *daf-2* exon 12–17 was amplified from *daf-2a* cDNA using primers that introduced a 5' overhang complementary to Exon 11.5 and a 3' overhang complementary to the *unc-54* 3'UTR (Fragment 2, primers 53 and 54). pPD49.26 *daf-2p::daf-2b* cDNA was digested with AgeI and KpnI to generate the vector backbone for HiFi DNA assembly with Fragments 1 and 2. The resulting plasmid pPD49.26 *daf-2p::daf-2c* was sequenced

across the entire *daf-2c* insert to confirm the correct sequence. To insert the gene of interest into a plasmid containing the positive selection marker, pCFJ151 was cut with SbfI and AvrII and *daf-2p:: daf-2b cDNA::unc-54 3'UTR* and *daf-2p::daf-2c cDNA::unc-54 3'UTR* were cloned in as SbfI/SpeI fragments.

## *C. elegans* strains and maintenance

*C. elegans* strains were maintained as previously described (*Brenner, 1974*). Bristol N2 (wild-type), DR1572[*daf-2(e1368) III*], JT709[*pdk-1(sa709) X*], EG6699[*ttTi5605 II; unc-119(ed3) III; oxEx1578*], GE24[*pha-1(e2123) III*] were obtained from the *Caenorhabditis elegans* Genetics Center (University of Minnesota, MN).

## Transgenic strains

Strains expressing tdTomato splicing reporters for *daf-2b* or *daf-2a/c* were generated through microinjection. Each construct was injected at a concentration of 50 ng/μL with a coinjection marker concentration of 50 ng/μL. *jluEx131* was integrated via UV mutagenesis to generate two integrated lines *jluIs15* and *jluIs16*. To confirm tissue-specific expression of the splicing reporters, we co-injected *ges-1p::gfp* (10 ng/μl), *dpy-7p::gfp* (10 ng/μl) and *rgef-1p::gfp* (20 ng/μl) with the *daf-2a/c* reporter (50 ng/μl) and a coinjection marker (50 ng/μl). For *daf-2b* we injected *ges-1p::gfp* (50 ng/μl), *dpy-7p::gfp* (25 ng/μl), *rgef-1p::gfp* (50 ng/μl) and *unc-122p::gfp* (50 ng/μl) into the integrated *daf-2b* strain *jluIs15*.

Strains overexpressing *daf-2b* cDNA were generated by microinjection. Each construct was injected at a concentration of 25 ng/μL with a coinjection marker concentration of 5 ng/μL. *jluEx141* expressing neuronal DAF-2B in the *daf-2* background (strain MGL275) was integrated via mutagenesis to yield the integrated line *jluIs17*. Strains overexpressing *daf-28p::daf-28*, *rgef-1p::ins-6* and *ins-18p::ins-18* were generated by injecting each construct at a concentration of 10 ng/μL. Each insulin peptide transgenic line was crossed into the integrated neuronal DAF-2B OE line, *jluIs17*. Wild type and mutant *rab-3p::daf-2b::FLAG* were injected at a concentration of 25 ng/ μL with a coinjection marker concentration of 3–5 ng/μL.

## *daf-2b::mScarlet* Knock-in

CRISPR with homology directed repair (HDR) was used to knock-in the mScarlet fluorescent protein into the *daf-2* genomic locus. crRNA targeting a PAM sequence at the end of the intronic sequence retained in *daf-2b,* just before the in-frame stop codon, was identified using www.crispr.mit.edu (oligo 55). An mScarlet repair template was amplified from plasmid pBG-GY837 using primers 56 and 57 that introduced 35 bp homology regions and gel purified using the Qiagen Minelute Gel Extraction kit. The use of a TAA stop codon at the terminus of mScarlet disrupted the PAM site.

We used the co-CRISPR approach of Paix et al, in which the *dpy-10* locus is edited, using a *dpy-10* crRNA (oligo 58) and a *dpy-10* HDR template (oligo 59), synthesized as a single stranded oligo and resuspended at a concentration of 1 μg/μl. crRNA for *dpy-10* and *daf-2b*, as well as tracrRNA (oligo 60) were syntheisized by Dharmacon. Recombinant Cas9::NLS was purified from *E. coli* according to the method of *Paix et al. (2015)*.

Injection mixes were prepared according to *Paix et al. (2015)* with small modifications. 30 animals were injected and singled onto individual plates 25˚C. On the first day of adulthood, jackpot plates were identified and 16 animals were singled to new plates. F1 animals were genotyped for the presence of the mScarlet insertion and progeny were cloned out to identify mScarlet homozygotes (primers 61 and 62). Two independent mScarlet knock-in lines were identified (MGL367 and MGL368) and sequenced for correct insertion of mScarlet (primers 63–65). To determine the tissue-specific expression of DAF-2B::mScarlet, MGL367 was used to generate transgenic lines expressing *rgef-1p::GFP* (neurons), *unc-122p::GFP* (coelomocytes), *dpy-7p::GFP* (hypodermis) and *ges-1p::GFP* (intestine) by microinjection.

## *daf-2bc* deletion

CRISPR with HDR was used to generate a *daf-2bc* deletion (*daf-2bc(Δ))*. crRNA targeting PAM sequences in Exon 11 and exon 12 of *daf-2* were identified using www.crispr.mit.edu (oligos 66 and 67) A 125 bp single-stranded homology-directed repair template was designed with a 34 bp left

hand homology region and a 33 bp right hand homology region (oligo 68). 58 bp of intervening sequence consisted of the last 5 bp of exon 11 and the first 53 bp of exon 12. An XhoI site was introduced at the exon 11 PAM sequence and the exon 12 PAM sequence was inactivated by silent mutagenesis. The repair oligo was synthesized (Eurofins Genomics) and resuspended at a concentration of 2 µg/µl.

We used the co-CRISPR approach of Ward, in which the *pha-1(e2123)* temperature-sensitive lethal mutation is corrected by homology-directed repair (*Ward, 2015*), using a *pha-1* crRNA (oligo 69) and a *pha-1* HDR template (oligo 70), synthesized as a single stranded oligo and resuspended at a concentration of 1 µg/µl. crRNA for *pha-1*, *daf-2* exon 11 and exon 12 were synthesized by Dharmacon. *pha-1* mutant worms were maintained at 15°C on HB101 bacteria. Injection mixes were prepared according to *Paix et al. (2015)* with small modifications. 30 animals were injected and maintained as pools of 5 per plate at 25°C. On day two post-injection, viable L1 larvae were singled to new plates and maintained at 25°C. Rescued adult F1s were allowed to lay eggs at 25°C and then were PCR genotyped for the presence of the *daf-2bc* deletion (primers 71–73). The progeny of heterozygotes were singled and tested for homozygosity, after which homozygotes were sequenced across the exon 11/exon 12 boundary to ensure that HDR had proceeded correctly (primers 74 and 75). A homozygous mutant line, *daf-2(jlu1)* with the correct deletion sequence was selected and backcrossed 3x to wild type prior to further analysis.

## Single copy insertions of *daf-2b* and *daf-2c* in *daf-2bc(∆)* deletion background

MosSCI on LGII was carried out using the strain EG6699[*ttTi5605 II; unc-119(ed3) III; oxEx1578*] (*Frøkjaer-Jensen et al., 2008*) to insert single copies of *daf-2b* and *daf-2c* into the genome. EG6699 was maintained on HB101 plates by picking non-Unc animals and Unc animals were picked for injection. pCFJ151(*daf-2b*) or pCFJ151(*daf-2c*) were injected at 10 ng/µl, along with pCFJ601 (*eft-3p*::transposase) 50 ng/µL, pMA122 (*hspp::peel-1*) 10 ng/µL, pGH8 (*rab-3p*::mcherry) 10 ng/µL, pCFJ90 (*myo-2p*::mcherry) 2.5 ng/µL, pCFJ104 (*myo-3p*::mcherry) 5 ng/µL. Non-unc animals that did not express mCherry were singled and genotyped for the presence of the insertion (primers 76 and 77). Homozygotes were identified using primers 77–79. PCR was used to confirm that the full length construct had been integrated and the correct insertion location was determined using primers 78 and 79. Two independent lines were obtained each for *daf-2b* and *daf-2c*. Integrants were backcrossed 3x to wild type prior to further analysis.

## Genetic crosses

Transgenic animals were crossed into different genetic backgrounds by standard methods. The temperature sensitive Daf-c phenotype at 25°C was used to confirm the presence of *daf-2(e1368)*. Since *pdk-1* is on the X chromosome, we crossed hemizygous F1 males from a cross back into the *pdk-1* parental line to ensure *pdk-1* homozygosity before segregating heterozygotes for the gene of interest.

## RNA extraction and generation of cDNA by RT-PCR

Synchronous populations of N2 eggs were generated by hypochlorite treatment and deposited onto NGM plates seeded with OP50. Animals were harvested at 12 hr (L1), 24 hr (L2), 36 hr (L3), 48 hr (L4), 60 hr (young adults). Mixed embryos were generated by harvesting eggs without growth on *E. coli* OP50. Animals were washed from growth plates with S-basal into 1.5 mL non-stick Eppendorf tubes and washed at least five times with S-basal to remove contaminating bacteria. Animals were then resuspended in 300 µL 10 mM Tris 1 mM EDTA solution and placed on ice. Samples were homogenized by sonication (10 s max, 4–5 times) using a 130 Watt 20 kHz Ultrasonic Processor and 2 mm stepped microtip. Samples were retained on ice for a maximum of 30 min.

Total RNA was extracted using RNAzol RT (Catalogue: R4533 and Lot: MKCF8526) from Sigma-Aldrich with slight modifications to the manufacturer guidelines. To ensure that no contaminating gDNA exists within the sample, RNA was resuspended in nuclease free water followed by DNAse treatment (Invitrogen, catalogue:18068–015) for 1 hr at 37°C. After this, RNA was reprecipitated from samples using 4M LiCl for 1 hr at room temperature followed by centrifugation and ethanol wash.

RNA purity and yield was calculated using a nanodrop spectrophotometer. Samples that had good purity and yield were retained for RT-PCR by using 1 µg of total RNA to generate cDNA from polyadenylated transcripts using the Invitrogen SuperScript IV RT kit and oligoDT reverse primer (Invitrogen, catalogue 18091050). Samples were diluted 1:4 in 10 mM Tris 1 mM EDTA. Full length cDNA was amplified using primers 80 and 81, and the exon 11/3'UTR fragment was amplified using primers 82 and 83. To detect cDNA fragments specific to *daf-2a*, *daf-2b* and *daf-2c* from the same sample we used multiplex PCR (40 cycles) using a common exon 11 forward primer (primer 82), an exon 12 reverse primer for *daf-2a/daf-2c* (primer 73) and a *daf-2b*-specific primer (primer 81). To quantify the ratio of *daf-2a* to *daf-2b,* we performed multiplex PCR at a cycle number within the linear range of amplification (35 cycles) and used Image J to quantify the band intensities on the gel.

## Immunoprecipitation (IP) and CoIP

Coimmunoprecipitation experiments were performed as previously described (*Sharma et al., 2014*), with modifications. $1 \times 10^6$ HEK 293T/17 cells were seeded in 10 cm dishes and transfected 24 hr later with a mixture of 10 µg total DNA and jetPRIME transfection reagent (Polyplus), according to the manufacturers protocol. 5 µg each plasmid was used for pairwise transfections and for single plasmid transfections 5 µg pBluescript was included. 36–48 hr after transfection, cells were lysed with 1.0% Nonidet P-40 buffer (50 mM Tris, pH 7.5, 150 mM NaCl, 10% glycerol, 1 mM DTT, EDTA-free protease inhibitor pellets (Roche Applied Science), pepstatin, microcystin, $NaVO_4$, NaF, sodium molybdate, and β-glycerophosphate). 1000 µg of total protein from transfected 293 cells was used for individual coIP experiments. Lysates were incubated with primary antibody for 30 min (rabbit polyclonal anti-HA antibody, cat# 715500, Life Technologies) and precipitated for 4 hr with 10 µl of protein G agarose (Sigma, cat# 11719416001) at 4°C. Precipitates were boiled in SDS Laemmli sample buffer (BioRad) and run on a 4–12% Bis Tris acetate gel (Invitrogen). Gels were transferred to PVDF membranes in Tris acetate transfer buffer and immunoblotted (anti-Myc mouse monoclonal antibody 9B11 – cat# 2276S, anti-HA mouse monoclonal antibody – cat# 2367S). Blots were visualized with HRP-conjugated anti-mouse secondary antibody (Fisher cat# 9491974), enhanced chemiluminescent reagent (West Femto, Pierce), and x-ray film.

## Pheromone extract preparation

Dauer pheromone extract was prepared as previously described (*Karp, 2018*) with modifications. N2 worms harvested from 6 to 8 medium plates were inoculated into 4 2L flasks containing 200 mL S-medium + 20 µg/mL nystatin and 25 mL 20x concentrated *E. coli* OP50 and incubated in Innova 44 Incubator Shaker at 150 RPM and 23.5°C for 1 week. Flasks were removed every 24–48 hr to monitor growth. Flasks were reinoculated with 25 mL *E. coli* OP50 and fresh nystatin powder (5 mg per flask) after 1 week or when the initial food was cleared. Worms were grown for an additional week until cultures took on an oily, yellow-brown appearance.

To generate the crude pheromone extract, the conditioned growth media was collected into a 1L cylindrical flask and placed overnight in a 4°C incubator to allow gravity settling of worms and debris, before the supernatant was removed and centrifuged at 7,000 RPM for 15 min to remove remaining worms, debris, and bacteria. The pale-yellow liquid was then transferred to a 2L beaker onto a 60°C hotplate in a fume hood and allowed to evaporate until approximately 50 mL of a brown slurry remained. This slurry was transferred to a ceramic mortar to evaporate the rest of the solution, leaving behind a thick brown sludge. This sludge was kept slightly wet with water (roughly 1 mL) and mixed with the pestle. Pheromones were extracted in 25 mL of anhydrous ethanol by using the pestle to slowly release the pheromones into solution. The ethanol was recovered after approximately 10 mins of gentle stirring and the extraction procedure was repeated 5–7 times until the ethanol solution being pulled off was mostly clear. Afterwards, the combined ethanol solution containing the dauer pheromones was placed under a nitrogen evaporator until about 10 mL of the solution remained. This final solution was passed through a 0.2 µm PES filter and distributed to 1.5 mL Eppendorf tubes and kept at −20°C until needed.

## Dauer entry assays with dauer pheromone

Dauer pheromones extracted in ethanol were mixed into molten NGM without added peptones at an empirically derived dosage that is dependent on the extraction efficiency and concentration of

pheromones from batch to batch (generally 300–500 µL per 100 mL molten agar (v/v) yielding plates that do not exceed 0.5% ethanol). Peptone-free NGM plates with added pheromone extract were seeded with 200 µL of an overnight culture of *E. coli* OP50 resuspended in S-basal and 1 mg/mL ampicillin to prevent growth and allowed to dry in a sterile hood. Bacterial concentration was adjusted to $3 \times 10^9$ colony-forming-units/mL S-basal. 10 gravid hermaphrodite N2 animals at the first day of adulthood were placed onto the bacterial lawn of each analysis plate per biological replicate per condition for 2 hr followed by removal. Plates were then maintained at 25°C for 44 hr. Dauers were scored on the basis of morphology and expressed as a percentage of the population.

## Dauer entry assays with insulin-signaling mutants

35 mm diameter petri dishes containing peptone-free NGM media were seeded with 200 µL of an overnight culture of *E. coli* OP50 resuspended in S-basal ($5 \times 10^9$ colony-forming-units/mL) and 1 mg/mL ampicillin to prevent growth and allowed to dry in a sterile hood. 10 gravid hermaphrodite animals at the first day of adulthood were placed onto the bacterial lawn of each analysis plate per biological replicate per condition for 2 hr followed by removal. Plates were maintained at 23.2°C for 44 hr (*daf-2(e1368)*) or 26.8°C for 40 hr (*pdk-1(sa709)*). Dauers were scored on the basis of morphology and expressed as a percentage of the population. For transgenic lines, usually three but at least two independent isolates were utilized and compiled together. Entire experiments were replicated at least twice.

## Dauer recovery assays

To measure dauer recovery, dauer larva were generated by incubation at 25°C (*daf-2(e1368)*) or 27°C for animals bearing the *pdk-1(sa709)* allele in the manner described above for dauer entry conditions. Dauers were collected in ddH20 solution and washed once. Equal volume 2% SDS solution was added to worm populations in water and animals were incubated at room temperature for 45 min with periodic gentle mixing by tube inversion. After SDS treatment, populations were washed twice with ddH20 and worms (living dauers and dead carcasses) were placed onto the center of an unseeded cholesterol-free, peptone-free 1.5% agarose in S-basal plate. After 1 hr SDS-resistant dauers were recovered and 30–40 animals were placed in the center of a fresh 1.5% agarose plate seeded with *E. coli* OP50 ($5 \times 10^9$ colony forming units/mL). Recovery plates were maintained at 20°C except where noted and scored every 12 hr for the presence of non-dauers, indicating exit from the dauer state. Animals that never recover from dauer arrest during the observation window were right-censored. For transgenic lines, usually three but at least two independent isolates were utilized and compiled together. Entire experiments were replicated at least twice.

## Lifespan assays

Lifespan assays were performed at 20°C on *E. coli* OP50 bacteria on NGM agar plates with a fresh lawn of bacteria. L4 larvae from a synchronized lay were transferred to a fresh plate and transferred daily during the reproductive period to prevent progeny contamination. Death was scored by loss of touch-provoked movement and animals lost due to bagging, uteral prolapse, or crawling up the side of the petri dish were censored. For transgenic lines, usually three but at least two independent isolates were utilized and compiled together. Entire experiments were replicated at least twice. Lifespan data were graphed using a Kaplan-Meier format and analyzed using the Log-Rank Test.

## Fluorescence imaging

Epifluorescence animal imaging was performed on a ZEISS Axio Observer A1 inverted microscope affixed with objective lenses ranging from 10 to 100x magnification.

Confocal microscopy was performed in the Light Microscopy Core at the Max Planck Florida Institute for Neuroscience. Worms were mounted live on 2% agarose pad, anesthetized with 10% $NaN_3$ and imaged using an LSM Zeiss 780 confocal microscope. The Z-stack images were acquired at 1 µm slice intervals at 63X. GFP and mScarlet excitation/emission were set to 488/526 and 651/632 respectively and each laser was in an independent track. L1 stage and young adult were imaged 20 hr and 72 hr after egg-layer respectively. Dauers were generated as previously described and imaged after SDS selection.

## Statistical analysis

The sample size for each experiment was determined empirically and was based on accepted practice within the *C. elegans* field. Statistical analysis was performed using GraphPad Prism v 8.0 with $p < 0.05$ indicating significance. For pairwise comparisons we used the Student's *t*-test without correction. For comparisons k > 2, One-Way ANOVA followed by a Tukey's *post hoc* test was used. The Log-Rank test was used to analyze dauer recovery data.

## Acknowledgements

We thank members of the Grill Lab and Roy Smith for useful discussions. This study was supported by the National Institutes of Health (NIH) grants AG050172 and DK108801. Some strains were provided by the *Caenorhabditis* Genetics Center, which is funded by the NIH Office of Research Infrastructure Programs (P40 OD010440). We thank Dr Long Yan, Head of Light Microscopy at the Max Planck Florida Institute of Neuroscience, for assistance with confocal microscopy.

## Additional information

### Funding

| Funder | Grant reference number | Author |
| --- | --- | --- |
| National Institutes of Health | AG050172 | Matthew S Gill |
| National Institutes of Health | DK108801 | Matthew S Gill |

The funders had no role in study design, data collection and interpretation, or the decision to submit the work for publication.

### Author contributions

Bryan A Martinez, Conceptualization, Formal analysis, Investigation, Writing - review and editing; Pedro Reis Rodrigues, Prosenjit Mondal, Aditi U Gurkar, Conceptualization, Investigation, Writing - review and editing; Ricardo M Nuñez Medina, Neale J Harrison, Investigation, Writing - review and editing; Museer A Lone, Methodology, Writing - review and editing; Amanda Webster, Investigation; Brock Grill, Conceptualization, Writing - review and editing; Matthew S Gill, Conceptualization, Supervision, Funding acquisition, Investigation, Methodology, Writing - original draft, Writing - review and editing

### Author ORCIDs

Neale J Harrison [ID] https://orcid.org/0000-0001-6821-4089
Brock Grill [ID] http://orcid.org/0000-0002-0379-3267
Matthew S Gill [ID] https://orcid.org/0000-0003-0818-8792

### Decision letter and Author response

Decision letter https://doi.org/10.7554/eLife.49917.sa1
Author response https://doi.org/10.7554/eLife.49917.sa2

## Additional files

### Supplementary files

- Supplementary file 1. Summary data for all replicates of lifespan experiments.
- Supplementary file 2. List of primers used in this study.
- Supplementary file 3. List of plasmids used in this study.
- Supplementary file 4. List of *C. elegans* strains used in this study.
- Transparent reporting form

## Data availability

All data generated or analysed during this study are included in the manuscript and supporting files.

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
