## [Decision Letter]

**Acceptance summary:**

Insulin signaling plays a critical role in regulating development and in neuromodulation in *C. elegans*. The *C. elegans* genome encodes ~40 insulin-like peptides (ILPs) which appear to signal via the single DAF-2 insulin receptor. Multiple DAF-2 isoforms generated via alternative splicing have been described previously and exhibit distinct functions and localization patterns. In this work, the authors describe a novel non-signaling DAF-2b isoform that retains the ligand-binding and transmembrane but not the intracellular signaling domain, and that may be secreted. This isoform can homodimerize and heterodimerize with the full-length DAF-2A protein and appears to modulate insulin signaling via sequestration of both agonistic and antagonistic ILPs. The authors also demonstrate active developmental stage-specific regulation of expression of the *daf-2b* isoform. This work describes a new mechanism by which the single DAF-2 receptor fine tunes ILP signaling as a function of developmental stage and conditions in the nematode.

**Decision letter after peer review:**

Thank you for submitting your article "An alternatively spliced insulin receptor modulates insulin sensitivity via insulin peptide sequestration in *C. elegans*" for consideration by *eLife*. Your article has been reviewed by three peer reviewers, and the evaluation has been overseen by Piali Sengupta as Reviewing Editor and Utpal Banerjee as the Senior Editor. The reviewers have opted to remain anonymous.

The reviewers have discussed the reviews with one another and the Reviewing Editor has drafted this decision to help you prepare a revised submission.

Summary:

Given the critical role of insulin signaling via the DAF-2 receptor in the regulation of multiple aspects of *C. elegans* behavior and development, all reviewers appreciated the importance of identifying the *daf-2b* alternatively spliced isoform. However, there were several issues raised that will need to be addressed in a revised manuscript.

Essential revisions:

1) Given that this is the first description of the *daf-2b* isoform, all reviewers agreed that its expression pattern needs to be described in greater detail and at higher resolution.

Specifically:

a) Please address how abundant this isoform is relative to *daf-2A/daf-2C*. Generally, this could be achieved by designing a primer mixture to multiplex/amplify/detect all three isoforms simultaneously on the gel (each isoform amplicon should be resolved on an agarose gel from the same reaction). Although semi-quantitative, this would give a better impression of how abundant the B isoform is relative to the other two species.

The authors should already have the RNA collected to do this experiment across developmental stages. It would also be interesting to collect RNA from dauer animals as a comparison as this is a key developmental stage discussed throughout the manuscript.

b) The fluorescence micrographs are of insufficiently high resolution. The authors should take higher magnifications of different anatomical regions of transgenic animals in order to give a better impression of the results presented.

Moreover, it is difficult to clearly ascertain cell types objectively from the micrographs given how zoomed out they are (especially the most important Figure 2D). The cell/tissue types need to be definitively identified.

Since these are tdTomato transgenes, the authors should obtain available GFP marker strains for neurons and hypodermis (intestine would be good as well but this is a bit more obvious to see even at lower magnification), and present co-localization data with cell types of interest at higher magnification.

2) "Most of the *C. elegans* insulin peptides are thought to act as agonists" is not quite correct, as the authors acknowledge in the last section of the Results. Many act in agonist/antagonist pairs (e.g., *ins-7* and *ins-18*, Murphy et al., 2003, Shaw et al., 2007; Fernandez de Abreu et al., 2014) and were recently well characterized as agonists and antagonists (Zheng et al., 2018). Given that there are both agonists and antagonists, how do the authors propose that antagonists are affected by the DAF-2B isoform? The authors have somewhat touched on this topic indirectly in Figure 7 using dauer pheromone, but with no details that directly test an antagonistic insulin. This part of the model should be better addressed, as it is hard to reconcile a "sopping up" model with antagonist insulin-like peptides.

3) Much of the data in the paper are derived from overexpressed transgenes from extrachromosomal arrays. While this is reasonable in some cases, the main conclusions do need to be confirmed using endogenously tagged proteins. Specifically, the recommendation is to generate splicing reporters using CRISPR knockins to demonstrate spatiotemporal regulation of production of the *daf-2b* isoform from the endogenous locus. This would also allow you to confirm that DAF-2b forms homodimers and heterodimerizes with DAF-2A via co-IP of endogenously tagged proteins in vivo from the animal. An alternative would be to create mosSCI/miniMos insertions with endogenous promoter, all exons/introns and native UTRs.

[Editors' note: further revisions were suggested prior to acceptance, as described below.]

Thank you for submitting your article "An alternatively spliced insulin receptor modulates insulin sensitivity via insulin peptide sequestration in *C. elegans*" for consideration by *eLife*. Your article was re-reviewed by one peer reviewer, and the evaluation has been overseen by a Reviewing Editor and Utpal Banerjee as the Senior Editor.

We agree that the revised manuscript has been substantially improved by the addition of new data. However, some of the data require additional revisions before the paper can be formally accepted.

For the microscopy data, the images could be further improved:

1) All images should include scale bars.

2) If available, bright field / DIC images or equivalent would be helpful for readers who are non-*C. elegans* experts to orient themselves, because the images are all at different scales (ranging from whole animal to single cell).

3) For the multiplex RT-PCR data:

Although these are semi-quantitative in nature, it would have been preferable/better if the authors performed a titration to ensure they were in the linear range of amplification / detection.

For example, in the present gel image, the *daf-2A* isoform appears to be saturating at this cycle number and RNA input amount, so it's hard to tell if the *daf-2A* isoform is 10x more abundant or 100x more abundant than the 2B isoform.

Generally, the standard would be to provide quantification as a mean + SD of the biological replicates.

4) With regards to trying to reconcile the endogenous protein expression vs. the splicing reporter expression data:

The authors provide one interpretation for the discrepancy of the two experimental datasets. However, it is possible that the *daf-2* promoter fragment / *unc-54* UTR combination used in the splicing reporter is missing regulatory information or causing spurious expression that is distinct from the endogenous protein expression pattern generated by the CRISPR fusion. This additional possibility should be discussed.

---

## [Author Response]

Essential revisions:1) Given that this is the first description of the daf-2b isoform, all reviewers agreed that its expression pattern needs to be described in greater detail and at higher resolution.Specifically:a) Please address how abundant this isoform is relative to daf-2A/daf-2C. Generally, this could be achieved by designing a primer mixture to multiplex/amplify/detect all three isoforms simultaneously on the gel (each isoform amplicon should be resolved on an agarose gel from the same reaction). Although semi-quantitative, this would give a better impression of how abundant the B isoform is relative to the other two species.The authors should already have the RNA collected to do this experiment across developmental stages. It would also be interesting to collect RNA from dauer animals as a comparison as this is a key developmental stage discussed throughout the manuscript.

We have performed the experiment as suggested. We generated new RNA samples in triplicate for L1-YA and in duplicate for dauer animals. We used the suggested multiplex approach, with a common forward primer in exon 11, a reverse primer in exon 12 which amplifies both *daf-2a* (568bp) and *daf-2c* (814bp), and a primer specific to the *daf-2b* transcript (500bp product). All samples were amplified at the same time and run on the same gel, with the amplification performed on two separate occasions. As expected, the most abundant isoform was *daf-2a*, with *daf-2b* detected at lower abundance in all samples including dauer. We have included a representative gel from one of the replicates in Figure 1 and included the following text:

“To estimate the relative abundance of the *daf-2b* transcript relative to *daf-2a* and *daf-2c*, we performed multiplex PCR. […] The abundance of *daf-2b* is lower than *daf-2a*, but appears to be higher than *daf-2c* (Figure 1E).”

b) The fluorescence micrographs are of insufficiently high resolution. The authors should take higher magnifications of different anatomical regions of transgenic animals in order to give a better impression of the results presented.Moreover, it is difficult to clearly ascertain cell types objectively from the micrographs given how zoomed out they are (especially the most important Figure 2D). The cell/tissue types need to be definitively identified.Since these are tdTomato transgenes, the authors should obtain available GFP marker strains for neurons and hypodermis (intestine would be good as well but this is a bit more obvious to see even at lower magnification), and present co-localization data with cell types of interest at higher magnification.

We have retaken images at 40x magnification for the both the *daf-2a/c* and *daf-2b* splicing reporters and amended Figure 2 accordingly. In addition, we generated *daf-2a/c::tdTomato* and *daf-2b::tdTomato* lines with *ges-1p::gfp, dpy-7p::gfp*, and *rgef-1p::gfp* to confirm localization to the intestine, hypodermis and neurons, respectively. These data have been included as Figure 2—figure supplement 1 and Figure 2—figure supplement 2.

In performing these co-localization experiments, we were able to confirm the neuronal expression of *daf-2b::tdTomato* in dauer animals but were unable to confirm hypodermal expression. Likewise we did not detect hypodermal expression of *daf-2a/c::tdTomato* in dauers. This has been corrected in the manuscript.

2) "Most of the C. elegans insulin peptides are thought to act as agonists" is not quite correct, as the authors acknowledge in the last section of the Results. Many act in agonist/antagonist pairs (e.g., ins-7 and ins-18, Murphy et al., 2003, Shaw et al., 2007; Fernandez de Abreu et al., 2014) and were recently well characterized as agonists and antagonists (Zheng et al., 2018). Given that there are both agonists and antagonists, how do the authors propose that antagonists are affected by the DAF-2B isoform? The authors have somewhat touched on this topic indirectly in Figure 7 using dauer pheromone, but with no details that directly test an antagonistic insulin. This part of the model should be better addressed, as it is hard to reconcile a "sopping up" model with antagonist insulin-like peptides.

We have now generated INS-18 over-expression lines (using the native *ins-18* promoter) and found that overexpression of INS-18 enhances dauer formation in response to pheromone in wild type animals, indicating that INS-18 is acting as an antagonist in this context. In the presence of extra copies of DAF-2B, INS-18 is no longer able to enhance dauer entry, supporting the model that DAF-2B is also capable of sequestering insulin peptide antagonists. These data have been included in Figure 7F. The following section has been added to the Results:

“To test whether DAF-2B could sequester antagonist insulin peptides under these conditions, we generated INS-18 overexpressing animals and examined their ability to influence pheromone-induced dauer formation in the presence or absence of DAF-2B. When over-expressed alone, INS-18 enhanced dauer entry in the presence of pheromone, indicating antagonist activity (Figure 7F). […]

The question of how do insulin peptides act as antagonists and perhaps more intriguingly, how do some peptides act as both agonists and antagonists, depending on the context, has not been resolved. However, we favor a model in which DAF-2B is equally capable of sequestering an antagonistic peptide or an agonist as we show in the current manuscript. However, more work will be required to fully elucidate the mechanism, which may well be insulin specific.

We have also added a paragraph to the Discussion:

“Overexpression studies have indicated that a number of insulin peptides in *C. elegans* can have mixed function, in that they act as agonists or antagonists depending on the context [Zheng et al., 2018]. […] Thus, reduced INS-18 signaling due to DAF-2B sequestration could lead to an upregulation of INS-7 expression, thereby further suppressing dauer formation.”

3) Much of the data in the paper are derived from overexpressed transgenes from extrachromosomal arrays. While this is reasonable in some cases, the main conclusions do need to be confirmed using endogenously tagged proteins. Specifically, the recommendation is to generate splicing reporters using CRISPR knockins to demonstrate spatiotemporal regulation of production of the daf-2b isoform from the endogenous locus. This would also allow you to confirm that DAF-2b forms homodimers and heterodimerizes with DAF-2A via co-IP of endogenously tagged proteins in vivo from the animal. An alternative would be to create mosSCI/miniMos insertions with endogenous promoter, all exons/introns and native UTRs.

In order to confirm that DAF-2B is expressed from the endogenous locus, we performed CRISPR knockin of mScarlet to generate a C-terminal DAF-2B::mScarlet translational fusion. Specifically, we identified a PAM site just before the in-frame stop codon in the portion of the intron that is retained in DAF-2B. Using a crRNA targeting this PAM site and a PCR-derived homology directed repair template, we were able to generate 2 independent knockins of mScarlet into the *daf-2b* locus. Sequencing confirmed the correction insertion of mScarlet. Both lines showed identical expression patterns and thus we chose just one for co-localization experiments.

We used confocal microscopy to characterize the expression pattern of this endogenous DAF-2B translational fusion through development into adulthood, as well as in dauers. We also introduced extrachromosomal arrays into this background to confirm tissue localization. We were able to confirm some neuronal expression of DAF-2B::mScarlet, but we did not observe intestinal or hypodermal expression, even in L1 animals, which have the strongest expression of the DAF-2B:tdTomato splicing reporter in these tissues. However, we did observe DAF-2B::mScarlet in coelomocytes in all developmental stages, including dauers, which we confirmed by colocalization with an *unc-122p::gfp* coelomocyte marker. Accumulation of translational GFP fusions in coelomocytes is a hallmark of a secreted protein, as has been demonstrated for the insulin peptide DAF-28. We interpret these data as an indication that DAF-2B is likely secreted. In support of this, we introduced the *unc-122p::gfp* coelomocyte marker into the DAF-2B::tdTomato splicing reporter strain and failed to see colocalization, indicating that DAF-2B is not expressed in coelomocytes.

In light of these new observations, we have made some modifications to Figure 2. We now include confocal images that demonstrate expression of DAF-2B::mScarlet in the nervous system and the coelomocytes, with colocalization markers, in L1, young adults and dauers. With respect to neuronal expression, we do not see explicit colocalization of DAF-2B::mScarlet with *rgef-1p::gfp* which could be due to the cytosolic location of GFP versus a cell surface localization of DAF-2B::mScarlet. However the location of DAF-2B::mScarlet around the periphery of the GFP positive cells leads us to believe that DAF-2B is in neurons in these images. In terms of the coelomocyte expression, DAF-2B::mScarlet localizes to the vacuoles of the coelomocyte and is clearly within the boundaries of the coelomocyte. We have also included *unc-122p*::GFP/DAF-2B::tdTomato images to demonstrate that DAF-2B splicing does not occur in coelomocytes.

It was mentioned in the recommendation that generation of a genomic knockin for DAF-2B would also facilitate co-immunoprecipitation experiments with DAF-2A. While we agree with the reviewers that this would be interesting, we have not pursued this line of experiments in this revised manuscript. There are a number of methodological considerations that make this a technically challenging approach that would demand more time than was available in this revision period. However, we remain interested in this aspect of DAF-2B biology and thank the reviewers for pushing us toward generating this knockin. It will be a valuable reagent for future studies of DAF-2B biology.

Based on these data, we have included the following section in the Results:

“To establish whether DAF-2B protein is expressed in vivo, we used CRISPR-Cas9 editing to introduce the coding sequence for the mScarlet fluorescent protein into the *daf-2b* genomic locus immediately after exon 11.5 (Figure 2E). […] For example, this might explain why we observed expression of *daf-2b* splicing in intestine, but couldn’t detect DAF-2B:mScarlet in this tissue..”

The following section has been added to the Discussion:

“To confirm endogenous expression of DAF-2B protein, we used CRISPR/Cas9 gene editing to insert mScarlet into the *daf-2* genomic locus such that fluorescence would only be observed when DAF-2B protein is generated. […] The lack of DAF-2B::tdTomato fluorescence from the splicing reporter in coelomocytes indicates that they are not a site of expression of DAF-2B, and thus, the appearance of DAF-2B::mScarlet in these cells supports the idea that DAF-2B may act as a secreted protein.”

[Editors' note: further revisions were suggested prior to acceptance, as described below.]

We agree that the revised manuscript has been substantially improved by the addition of new data. However, some of the data require additional revisions before the paper can be formally accepted.For the microscopy data, the images could be further improved:1) All images should include scale bars.

All microscopy images now include scale bars.

2) If available, bright field / DIC images or equivalent would be helpful for readers who are non-C. elegans experts to orient themselves, because the images are all at different scales (ranging from whole animal to single cell).

We have included brightfield images for the confocal images in Figure 2 in the form of a brightfield/mscarlet/gfp merge image. We have also included bright field images for Figure 2—figure supplement 1 and Figure 2—figure supplement 2. In addition, we have modified the images to present mScarlet and tdTomato as magenta rather than red in order to accommodate readers with color blindness, as per the *eLife* figure guidelines.

3) For the multiplex RT-PCR data:Although these are semi-quantitative in nature, it would have been preferable/better if the authors performed a titration to ensure they were in the linear range of amplification / detection.For example, in the present gel image, the daf-2A isoform appears to be saturating at this cycle number and RNA input amount, so it's hard to tell if the daf-2A isoform is 10x more abundant or 100x more abundant than the 2B isoform.Generally, the standard would be to provide quantification as a mean + SD of the biological replicates.

Multiplex PCR. As suggested, we established the linear range of amplification by examining product accumulation between 33 and 43 cycles and found that the signal started to plateau at 37 cycles. For quantitation, we therefore used 35 cycles to amplify cDNA from our larval samples and ran the reactions on the same gel, using Image J to quantify band intensities. Using this approach, we found that *daf-2a* was between 4 and 7 times higher abundance than *daf-2b*. We have included these data as Figure 1—figure supplement 2 and we have included the following sentence in the Results:

“Analysis of stage-specific cDNA showed that *daf-2b* is present at all life stages (Figure 1E) and *daf-2a* is between 4 and 7 times more abundant than *daf-2b* (Figure 1—figure supplement 2).”

A legend for Figure 1—figure supplement 2 is included:

“Multiplex PCR was performed within the linear range of amplification and band intensities were measured using Image J. Each larval stage was examined in triplicate except dauer (duplicate) and data are presented as mean + sd.”

The gel image shown in Figure 1E was from multiplex PCR at 40 cycles and has been over-exposed which was necessary to show the presence of the *daf-2c* isoform. We have noted the cycle number for these data and as well as the quantification data in the Materials and methods section (subsection “RNA extraction and generation of cDNA by RT-PCR”, last paragraph).

4) With regards to trying to reconcile the endogenous protein expression vs. the splicing reporter expression data:The authors provide one interpretation for the discrepancy of the two experimental datasets. However, it is possible that the daf-2 promoter fragment / unc-54 UTR combination used in the splicing reporter is missing regulatory information or causing spurious expression that is distinct from the endogenous protein expression pattern generated by the CRISPR fusion. This additional possibility should be discussed.

We recognize the alternate explanation for the discrepancy between the endogenous protein expression and the splicing reporter and have included the following sentence in the Discussion:

“However, it should also be noted that differences in the pattern of expression between the endogenous fusion protein and the splicing reporter could arise from the absence of important regulatory sequences and / or the presence of the *unc-54* UTR in the splicing construct.”